# Transiently heritable fates and quorum sensing drive early IFN-I response dynamics

**Laura C Van Eyndhoven[1,2], Vincent PG Verberne[1,2], Carlijn VC Bouten[2,3], Abhyudai Singh[4], Jurjen Tel[1,2]\***

[1]Laboratory of Immunoengineering, Department of Biomedical Engineering, Eindhoven University of Technology, Eindhoven, Netherlands; [2]Institute for Complex Molecular Systems (ICMS), Eindhoven University of Technology, Eindhoven, Netherlands; [3]Department of Biomedical Engineering, Eindhoven University of Technology, Eindhoven, Netherlands; [4]Department of Electrical and Computer Engineering, University of Delaware, Newark, United States

**Abstract** Type I interferon (IFN-I)-mediated antiviral responses are central to host defense against viral infections. Crucial is the tight and well-orchestrated control of cellular decision-making leading to the production of IFN-Is. Innovative single-cell approaches revealed that the initiation of IFN-I production is limited to only fractions of 1–3% of the total population, both found in vitro, in vivo, and across cell types, which were thought to be stochastically regulated. To challenge this dogma, we addressed the influence of various stochastic and deterministic host-intrinsic factors on dictating early IFN-I responses, using a murine fibroblast reporter model. Epigenetic drugs influenced the percentage of responding cells. Next, with the classical Luria–Delbrück fluctuation test, we provided evidence for transient heritability driving responder fates, which was verified with mathematical modeling. Finally, while studying varying cell densities, we substantiated an important role for cell density in dictating responsiveness, similar to the phenomenon of quorum sensing. Together, this systems immunology approach opens up new avenues to progress the fundamental understanding on cellular decision-making during early IFN-I responses, which can be translated to other (immune) signaling systems.

**\*For correspondence:**
j.tel@tue.nl

**Competing interest:** The authors declare that no competing interests exist.

## Editor's evaluation

This important study combines quantitative experiments and modeling to dissect the factors guiding cell fate decisions during early antiviral (type I interferon) signaling. The authors provide solid evidence that the fate of cells is transiently heritable, and uncover a role for cell density in regulating responsiveness, reminiscent of a quorum sensing mechanism. This work will be of broad interest to systems biologists, immunologists, and cell biologists.

## Introduction

Type I interferon (IFN-I)-mediated responses are central to host defense against viral infections (*Ivashkiv and Donlin, 2014*; *Mesev et al., 2019*). Crucial is the tight and well-orchestrated control of cellular decision-making leading to the production of IFN-I, as impaired response dynamics leads to the pathogenesis of a plethora of diseases that go beyond antiviral immunity only (*Musella et al., 2017*; *Park and Iwasaki, 2020*; *Psarras et al., 2017*; *Zhang et al., 2020*). Over the past decades, multilayered stochasticity (i.e., stochasticity originating from distinct, but intertwined layers over

**eLife digest** When we start to develop a cold, influenza or another viral infection, some of our cells produce signaling molecules known as type I interferons (or IFN-Is for short). These early IFN-I signals establish defenses against viruses in both infected and as yet uninfected cells. If the cells produce too much IFN-Is, however, it can result in uncontrolled inflammation that may harm the body and cause life threatening illness.

Individual cells need to tightly control how much IFN-Is they produce and match this with the course of the viral infection. They also need to assess how much IFN-I their neighbors are producing and adjust their behavior accordingly.

Cells have evolved a myriad of mechanisms to ensure the right amounts of IFN-Is are produced in different circumstances. Broadly, these mechanisms can be divided into two categories: stochastic regulation and deterministic regulation. Stochastic regulation occurs when individual cells receive the exact same information, but this leads to different outcomes, such as, different cells producing various quantities of IFN-Is. In contrast, deterministic regulation causes the same outcome in different cells independent on the information they receive.

It was thought that stochastic regulation is the main driver of early IFN-1 responses, but recently a handful of studies have reported deterministic regulation being primarily responsible, instead. Here, Van Eyndhoven et al. explored the roles of both types of regulation in the early IFN-I responses of mouse cells.

Van Eyndhoven et al. used genetic approaches and mathematical modelling to show that the fraction of cells that initiate early IFN-I responses can be considered deterministic. Moreover, this deterministic feature turned out to be heritable, such that the fate to produce IFN-I gets passed on for several generations of cells. Additionally, the experiments suggest that cell density, that is, how tightly packed together the cells are, plays an important role in controlling how many cells make IFN-I, with a lower cell density resulting in a higher fraction of cells producing IFN-Is.

The findings of Van Eyndhoven et al. add to a growing body of evidence reporting heritable states that can guide decision-making in individual cells. Furthermore, it revises our view on how individual immune cells coordinate population-wide responses.

the course of an infection) driving cellular heterogeneity and subsequent cellular decision-making during IFN-I responses have become increasingly apparent (*Rand et al., 2012*; *Van Eyndhoven et al., 2021b*). In short, IFN-I responses are elicited by fractions of so-called first responding cells, also referred to as 'precocious cells' or 'early responding cells', which start the initial IFN-I production upon viral detection, both validated in vitro, in vivo, and across cell types (*Bauer et al., 2016*; *Hjorton et al., 2020*; *Patil et al., 2015*; *Shalek et al., 2014*; *Van Eyndhoven et al., 2021a*; *Wimmers et al., 2018*). Their IFN-I production is further enhanced via autocrine signaling, inducing a feedforward loop resulting in the upregulation of interferon regulatory factor (IRF) 7 and other signaling components (*Honda et al., 2006*). Simultaneously, first responders trigger additional IFN-I production in so-called second responders, which are activated upon IFN-mediated paracrine signaling in combination with viral detection (*Van Eyndhoven et al., 2021b*; *Wimmers et al., 2018*). These two major events have also been described as the early phase and later phase of IFN-I responses (*Honda et al., 2006*). Especially, the regulation of the early phase is of increasing interest, because this phase is currently thought to orchestrate population-wide IFN-I signaling, therefore of crucial importance in establishing systemic antiviral protection (*Patil et al., 2015*; *Van Eyndhoven et al., 2021b*).

Up till today it remains unclear whether cellular decision-making to become an IFN-I producer during the early phase is as a stochastic process (dictated by 'random' host-intrinsic factors, e.g., limiting levels of transcription factors and other signaling intermediates), or a deterministic process (dictated by 'predetermined' host-intrinsic factors, e.g., epigenetic profiles leading to a predispositioning to perform certain cellular behaviors). Importantly, although the terms stochasticity and determinism seem highly dichotomous, deterministic features (e.g., epigenetic regulation) are often, if not always, stochastically regulated (*Zernicka-Goetz and Huang, 2010*). However, in cellular decision-making, the major difference between a stochastic process and a deterministic process boils down to the effects of (varying) inputs on dictating (varying) outputs. In fact, a stochastic process in characterized

by the exact same stimulus leading to varying response outcomes, often as a result of varying host-intrinsic factors (*Symmons and Raj, 2016*). In contrast, a deterministic process is characterized by an outcome (e.g., IFN-I production) that is fixed, or at least to a large degree, while the input can be variable. Accordingly, the later IFN-I phase seems mainly driven by stochastic processes, as the outcome is highly heterogeneous, mainly dictated by limiting host-intrinsic factors (e.g., intrinsic and extrinsic gene expression noise), and matching the course of an infection (*Abadie et al., 2019*). The limiting host-intrinsic factors can be manipulated by overexpression of signaling intermediates, such as retinoic acid-inducible gene I (RIG-I), IRF3, and IRF7, leading to an increased overall production of IFN-Is (*Harrison and Moseley, 2020*; *Zhao et al., 2012*). In contrast, recent evidence suggests that the early phase, and more specifically the first responders, could be dictated by determinism instead (*Bagnall et al., 2020*; *Shaffer et al., 2020*; *Talemi and Höfer, 2018*; *Van Eyndhoven et al., 2021a*). That would imply that prior to an immune challenge, a dedicated subset of cells is epigenetically programmed to become a first responder, independent on the exact input. Accordingly, a transiently heritable gene expression program related to IFN-I signaling, including the expression of *RIG-I (DDX58)*, *IFIT1*, *PMAIP1*, and *OASL*, was discovered to be initiated only in fractions of unstimulated cells (*Shaffer et al., 2020*). Transient heritability refers to heritable epigenetic profiles (e.g., profiles encoding cellular fates for the production IFN-Is) that only transfer over a couple of generations, as observed across cell types and systems including cancer drug resistance (*Shaffer et al., 2020*; *Sharma et al., 2010*), cancer fitness (*Fennell et al., 2022*; *Oren et al., 2021*), NK cell memory (*Rückert et al., 2022*), HIV reactivation in T cells (*Lu et al., 2021*), epithelial immunity (*Clark et al., 2021*), and trained immunity (*Katzmarski et al., 2021*). Moreover, numerous studies characterized epigenetic control of IFN-I-related genes (e.g., *IFNB*), which may be of crucial importance during the early phase of IFN-I response dynamics (*Daman and Josefowicz, 2021*; *Gao et al., 2021*).

Besides a growing body of evidence on the role of transient heritable fates dictating cellular behaviors, the effects of population density, often referred to as quorum sensing, are getting more established for immune (signaling) systems (*Antonioli et al., 2019*; *Polonsky et al., 2018*; *Van Eyndhoven and Tel, 2022*). On top of the intrinsic features characterized by stochasticity and determinism, individual immune cells can communicate in various ways to elicit appropriate systemic immune responses. Typically, cytokine-mediated communication is categorized into two types: autocrine and paracrine signaling. Autocrine signaling is defined by cells secreting signaling molecules while simultaneously expressing the cognate receptor. Paracrine signaling is defined by cells either secreting signaling molecules without expressing the cognate receptor, or cells expressing the receptor without secreting the molecule. In essence, quorum sensing can be considered a phenomenon in which autocrine cells determine their population density based on cells engaging in neighbor communication, but without self-communication (*Doğaner et al., 2016*; *Van Eyndhoven and Tel, 2022*). Especially in the presence of other competitive decision makers (i.e., cytokine consumers and producers), it is critical for individual cells to assess cellular density, and act accordingly (*Oyler-Yaniv et al., 2017*). Given the risk of the pathogenesis of autoimmune diseases as a result of overshooting IFN-I production, we hypothesize that quorum sensing is controlling IFN-I response dynamics in a way in which the density of cells capable of IFN-I production is dictating the actual numbers of IFN-I producers, while similar phenomena are starting to get established for a wide variety of immune systems (*Antonioli et al., 2019*; *Bardou et al., 2021*; *Doğaner et al., 2016*; *Muldoon et al., 2020*; *Polonsky et al., 2018*; *Schrom et al., 2020*).

In this study, we addressed the influence of various stochastic and deterministic host-intrinsic factors on dictating early IFN-I responses in a murine fibroblast model (*Rand et al., 2012*). After having validated the fraction of first responders, which was remarkably similar to what has been observed and characterized in immune cells (e.g., plasmacytoid dendritic cells [DCs] and monocyte-derived DCs) and other cell systems (e.g., primary human fibroblasts and murine fibroblasts) before, we assessed the three most important aspects of extrinsic and host-intrinsic stochasticity on cellular decision-making (i.e., varying viral loads, heterogeneous IRF7 levels, and fluctuations in cell cycle states). Using epigenetic drugs and the classical Luria–Delbrück fluctuation test, we challenged the dogma on stochasticity dictating early IFN-I responses (*Wimmers et al., 2018*). Accordingly, our results provide additional proof on deterministic, transiently heritable fates driving responsiveness instead, similar to what has been observed for various other cell systems, which we substantiated with an ordinary differential equation (ODE) model (*Clark et al., 2021*; *Shaffer et al., 2020*). Finally, we assessed the effects of cell

density driving population-wide responsiveness, showing a robust effect of a low cell density resulting in higher responder percentages. Together, this systems immunology approach highlights the ability to revise the fundamentals of cellular decision-making during early IFN-I responses, and potentially other immune signaling systems. Ultimately, these novel insights pave the way toward improved IFN-mediated immune therapies.

## Results

### Reporter cell model to study early IFN-I responses

Studying IFN-I dynamics in (human) primary immune cells allows for translation toward clinical applications, however, experimental approaches are often limited by relatively low cell counts, possible immune cell impurities, and additional layers of stochasticity introduced by the presence of heterogeneous subsets (*Van Eyndhoven et al., 2021a*). Besides, a crucial role for structural cells as key regulators of organ-specific immune responses is getting increasingly recognized and established (*Krausgruber et al., 2020*). Therefore, we utilized murine reporter cells to provide us with a robust model to study early IFN-I responsiveness (*Rand et al., 2012*). In fact, fibroblast models, both human primary and murine, have broadly proven their suitability for studying antiviral immunity (*Drayman et al., 2019*; *Krausgruber et al., 2020*).

The early IFN-I phase is characterized by the detection of viral nucleic acids by pathogen recognition receptors, leading to the phosphorylation and translocation of IRFs (e.g., IRF3 and IRF7) from the cytoplasm to the nucleus, where they initiate the transcription of IFN-Is (*Honda et al., 2006*; *Rehwinkel and Gack, 2020*; *Figure 1A*). Subsequently, the later phase is characterized by the signaling induced by IFN-Is activating IFN-I receptors (IFNARs). This leads to the phosphorylation, complex formation, and translocation of signal transducer and activator of transcription 1 (STAT1), STAT2, and IRF9, termed IFN-stimulated gene factor 3 (ISGF3), to initiate the transcription of interferon-stimulated genes (ISGs). Accordingly, we used a NIH3T3:IRF7-CFP reporter cell line, expressing low, physiological background levels of IRF7-CFP fusion proteins, to monitor signaling dynamics during early phase IFN-I response dynamics (*Figure 1B*). For this cell model, IRF7 translocation correlates with IRF3 translocation, making IRF7 translocation as only readout sufficient to study first responders (*Rand et al., 2012*). The NIH3T3:STAT1-CFP/STAT2-YFP reporter cell line was utilized for validation of the production of IFN-Is upon translocation of IRF7.

To identify first responders, translocation IRF7-CFP fusion proteins were monitored in an unbiased fashion using a custom-made automated image analysis script developed in the CellProfiler software (*Figure 1C*; *Figure 1—figure supplement 1A–C*; *Figure 1—figure supplement 2A–D*; *Stirling et al., 2021*). Primary objects (nuclei) were detected and defined based on the Hoechst signal after nuclei staining. Next, the secondary objects (cells) were detected and defined based on the CFP signal originating from the IRF7-CFP fusion proteins molecules. Finally, the tertiary objects (cytoplasms) were defined by subtracting the primary objects from the secondary objects.

First responders could be defined by determining the IRF7 translocation ratio by dividing the CFP median intensity from the nucleus by the CFP median intensity from the cytoplasm (N/C) (see Materials and methods). As an example, six cells were imaged simultaneously, containing three responding cells showing clear translocation of signal, and three nonresponding cells showing relatively less signal inside the nucleus (*Figure 1D*). Indeed, the three cells that would have been defined by eye as responding cells had the highest IRF7 translocation ratio (*Figure 1E*).

Together, we established the detection of first responding cells in a high-throughput, unbiased manner, based on the translocation of fluorescent signal corresponding with IRF7 molecules from the cytoplasm to the nucleus.

### Validation of first responders in a reporter cell model

To elicit early IFN-I responses in our model, we used rhodamine-labeled Poly(I:C), instead of live or attenuated viruses, thereby avoiding any additional stochasticity introduced by viral extrinsic factors (e.g., genetic variability among the virus population, variability in viral replication, etc.). By using rhodamine-labeled Poly(I:C) over regular Poly(I:C), we were able to carefully track transfection efficiencies over time (*Figure 2A, B*). To limit noise introduced to the system, resulting from poor transfection

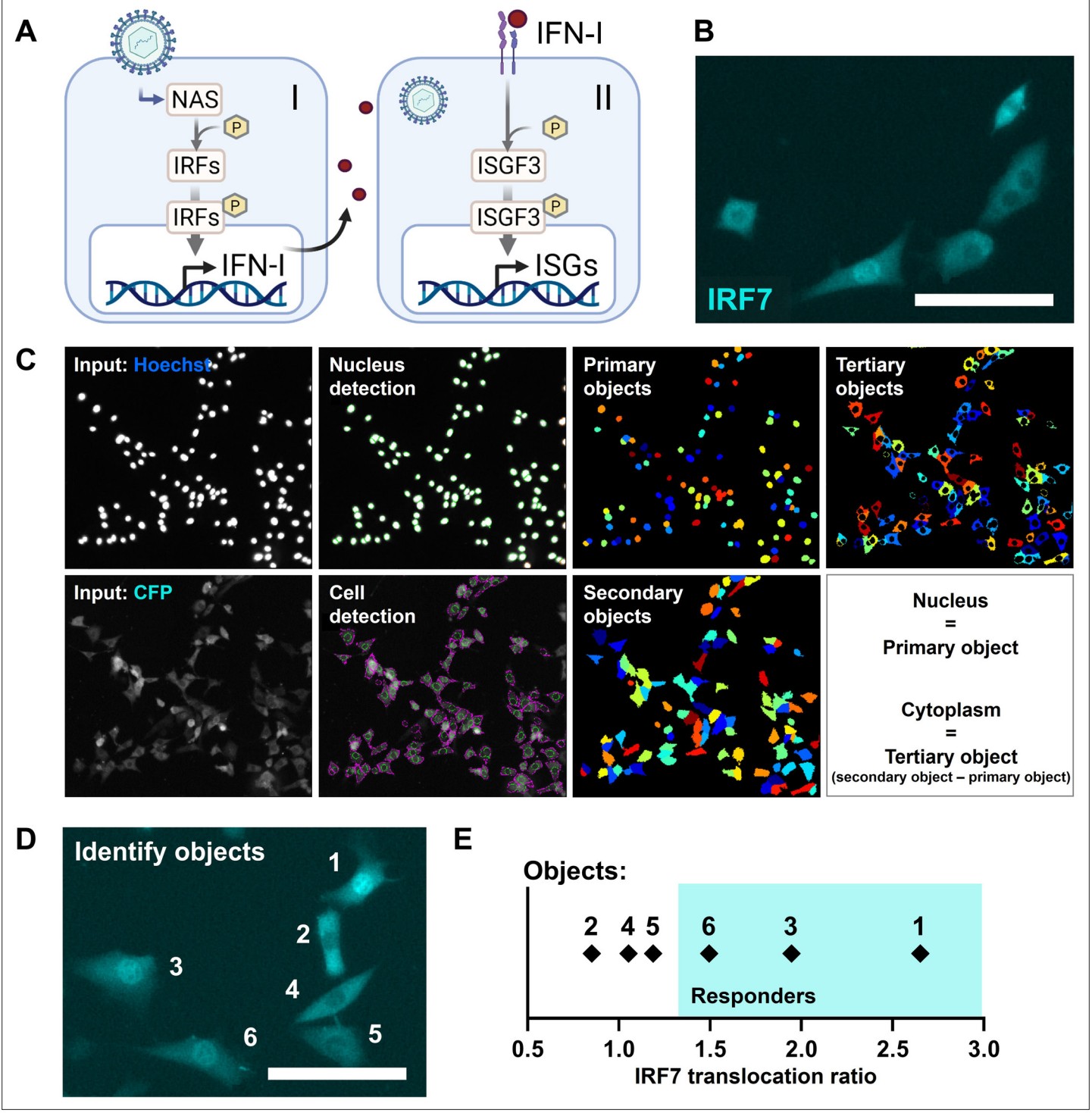

**Figure 1.** Reporter cell model to study early type I interferon (IFN-I) responses. (**A**) Schematic overview of the early (I) and later (II) phase IFN-I responses. The early phase is characterized by the detection of nucleic acids by nucleic acid receptors (NAS), followed by the phosphorylation (p) and translocation of interferon regulatory factors (IRFs) and subsequent induction of IFN-Is. Upon paracrine signaling, IFN-Is bind to IFN-I receptors (IFNARs), leading to the phosphorylation and translocation of interferon-stimulated gene factor 3 (ISGF3), consisting of STAT1, STAT2, and IRF9, respectively, inducing the production of interferon-simulated genes (ISGs). (**B**) Microscopy image of NIH3T3 cells stably expressing the fusion protein IRF7-CFP. Scale bar equals 100 μm. (**C**) Image processing and analysis steps in CellProfiler script for the detection of fluorescent signal in the nuclei and cytoplasms. (**D**) Example image with six identified objects. Scale bar equals 100 μm. (**E**) IRF7 translocation ratios of example objects plotted.

The online version of this article includes the following figure supplement(s) for figure 1:

*Figure 1 continued on next page*

*Figure 1 continued*

**Figure supplement 1.** Details on automated script in CellProfiler software.

**Figure supplement 2.** Detailed automated script analysis performance.

timing and efficiencies, we optimized transfection to achieve fast and potent delivery of stimulus (*Figure 2C*; *Figure 2—figure supplement 1A, B*).

Next, we set out to explore the response dynamics over the first 9-hr post transfection to determine the response peak (*Figure 2D*). Earlier studies indicated a peak of IRF7 translocation around 8 hr, and a peak of IFN-beta (IFNβ) production around 10-hr post activation (i.e., using Poly(I:C) and Newcastle Disease Virus) (*Rand et al., 2012*). Accordingly, upon transfection optimization, in our experiments the response peaked at 7-hr post transfection, with an average of 2.1% of responding cells (*Figure 2E, F*). This percentage is in line with what has been found across literature, species (i.e., human and mice) and cell types (i.e., fibroblasts, monocyte-derived DCs, plasmacytoid DCs), which ranges from 0.8 to 10% of early responders, emphasizing the elegant yet robust feature of only a fraction of first responding cells driving the population-wide IFN-I system (*Bauer et al., 2016*; *Drayman et al., 2019*; *Patil et al., 2015*; *Shalek et al., 2014*; *Van Eyndhoven et al., 2021a*; *Wimmers et al., 2018*). Besides, the background numbers of translocated cells possibly reflect the intrinsic feature of the IFN-I system to ensure basal IFN-I expression and IFNAR signaling to equip immune cells to rapidly mobilize effective antiviral immune responses, and homeostatic balance through tonic signaling (*Gough et al., 2012*; *Ivashkiv and Donlin, 2014*).

Accordingly, we wondered whether we could capture the orchestrating role of first responders on population-wide IFN-I response dynamics. Therefore, we studied the response dynamics using a NIH3T3:STAT1-CFP/STAT2-YFP fibroblast reporter cell line. Seven hours post infection, IFNβ produced by the first responders will diffuse to neighboring, yet nonresponding cells, thereby activating their IFNARs, followed by the subsequent translocation of ISGF3, consisting of STAT1, STAT2, and IRF9. Accordingly, at 7-hr post infection, we were able to capture clusters of STAT1/STAT2 translocated cells (*Figure 2G–I*). This timing is in agreement with earlier findings obtained using this cell line (*Rand et al., 2012*). The clusters represent a phenomenon of competition between cytokine diffusion (i.e., IFN-Is produced by first responder) and consumption (i.e., by surrounding cells) generating spatial niches of high cytokine concentrations with sharp boundaries (*Oyler-Yaniv et al., 2017*).

Taken together, we established a methodology for rapid and potent delivery of stimulus, thereby minimizing the potential noise introduced by extrinsic factors, to further reveal the multilayered stochasticity driving first responders. Additionally, we validated the presence of fractions of first responders, and validated their ability to induce population-wide IFN-I signaling.

## Extrinsic and intrinsic stochasticity dictating first responders

In contrast to the role of host-intrinsic factors, literature stated that the role of extrinsic factors (those that are introduced by the virus/stimulus itself) is rather small in determining the fraction of first responders, indicated by the lack of dose-dependent effects and the robustness of percentages of first responders across stimulus types (*Shalek et al., 2014*; *Van Eyndhoven et al., 2021a*; *Wimmers et al., 2018*). Of note, on the contrary, extrinsic factors can correlate with the percentage of second responders, though studies often do not distinguish between these two different cell fates, but focusing on population-wide responses instead (*Rand et al., 2012*; *Zhao et al., 2012*). To test the effect of a variety of extrinsic factors on the first responders, we first tested for a correlation between the responsiveness (i.e., IRF7 translocation ratio) and the actual amount of stimulus received by the cells, which was only very low, though significant ($R^2 = 0.0171$, $p < 0.0001$) (*Figure 3A*). While the events displaying the highest IRF7 translocation ratios (above 2) only displayed very low levels of Poly(I:C) mean intensities, we conclude that first responders are only minorly, if at all, influenced by stimulus dosage.

Like all biochemical reactions, stochastic processes (e.g., gene expression noise) influence IFN-I response dynamics. Universally, intrinsic gene expression noise results from the stochastic nature of biochemical reactions, whereas extrinsic gene expression noise results from cell–cell fluctuations of components that are involved in generating the response (*Dey et al., 2015*). In essence, every step of IFN-I signaling involves limiting signaling intermediates, making every step subject to the effects of gene expression noise (*Zhao et al., 2012*). While IRF7 is one of the key factors driving

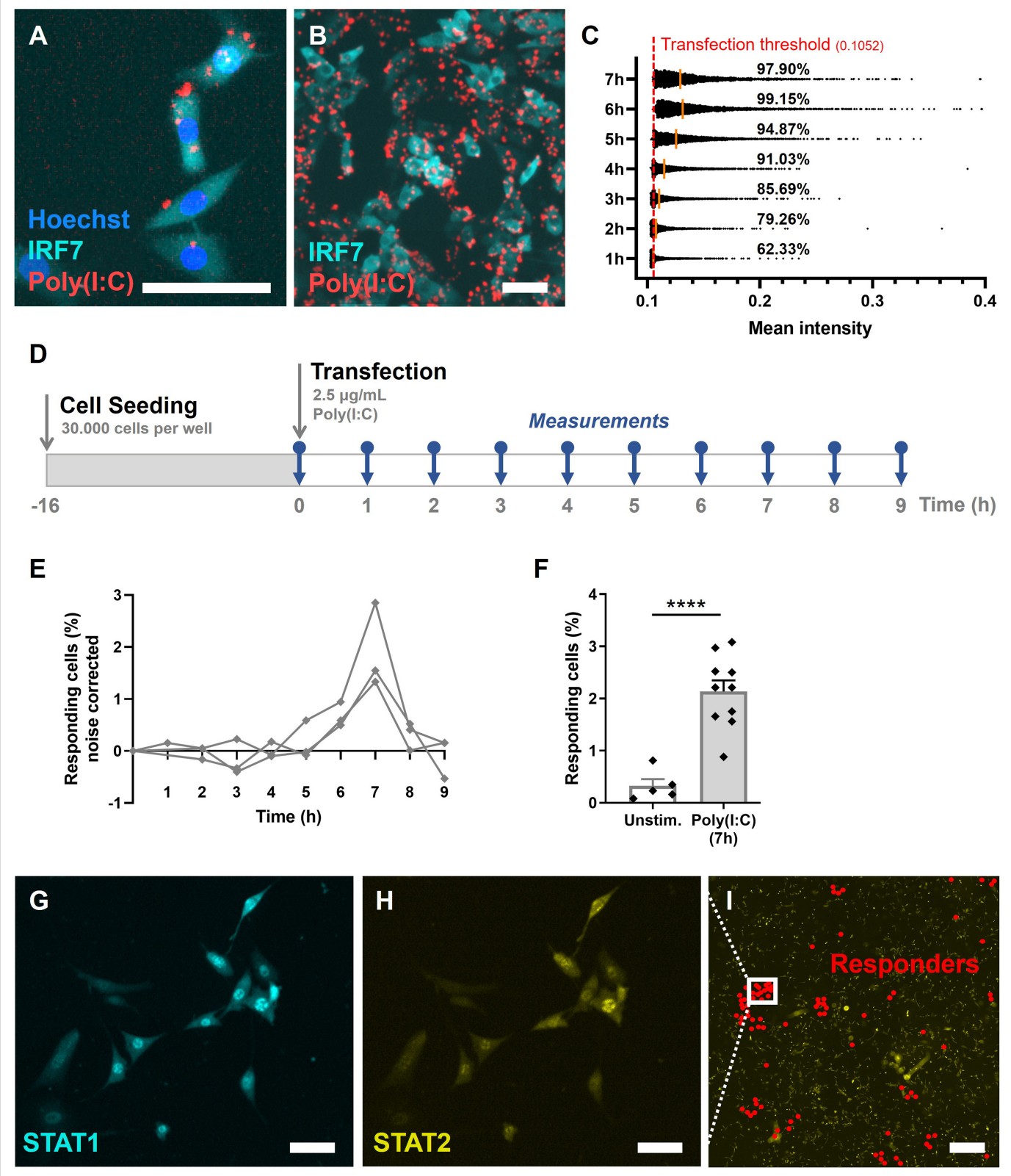

**Figure 2.** Validation of first responders in reporter cell model. (**A**) Microscopy picture of NIH3T3: IRF7-CFP, stained with Hoechst nuclear stain, transfected with rhodamine-labeled Poly(I:C). Scale bar equals 100 μm. (**B**) Overview of transfected cells. Scale bar equals 100 μm. (**C**) Transfection efficiency quantification over time, based on rhodamine mean intensity detected in cells. The red dotted line represents the transfection threshold. The orange lines represent the mean values. (**D**) Experimental design of first responder validation in NIH3T3: IRF7-CFP cells. (**E**) Percentages of noise

*Figure 2 continued on next page*

*Figure 2 continued*

corrected responding cells. Cells were seeded on coverslips 16 hr prior to transfection with 2.5 µg/ml Poly(I:C). Over the first 9 hr, the percentages of translocated cells were determined ($n = 3$ experimental replicates). (**F**) Percentages of responding cells after 7 hr of Poly(I:C) transfection, compared to unstimulated cells ($n = 10$; $p = 0.0003$); data are represented as mean ± standard error of the mean (SEM). ****$p \leq 0.001$ (Student's $t$-test). (**G**) Microscopy image of NIH3T3: STAT1-CFP; STAT2-YFP, for additional first responder validation. Cells were seeded and transfected as described before. Translocation of STAT1 was assessed after 7-hr post transfection. Scale bar equals 100 µm. (**H**) Corresponding image of STAT2-YFP signal. Scale bar equals 100 µm. (**I**) Corresponding overview image of population of NIH3T3: STAT1-CFP; STAT2-YFP, with responding (translocated) cells indicated with red dots. Scale bar equals 1 mm.

The online version of this article includes the following figure supplement(s) for figure 2:

**Figure supplement 1.** Confocal microscopy and flow cytometry analysis of transfection efficiency.

IFNβ production, and thereby possibly driving the first responders, we set out to investigate the relation between background levels of IRF7 and first responding cells. However, we could only find a very weak correlation between IRF7 translocation ratio and IRF7 mean intensity ($R^2 = 0.0620$, $p < 0.0001$), arguing that first responders are only minorly driven by differences in background levels of IRF7 (*Figure 3B*). Although, it was interesting to observe the degree of heterogeneity in background IRF7 expression levels, as well as the significant increase in signal after stimulation (*Figure 3C*). The latter confirms the already well-characterized feedback loops enhancing the IRF7 expression after autocrine and paracrine signaling induced by the first responders.

Next, we wondered whether cell cycle state could be a potential driver, since studies pointed toward a role for cell cycle state dictating IFN-I production, though mainly related to second responding cells (*Cingöz and Goff, 2018*; *Mudla et al., 2020*). To explore the effect of cell cycle state on first responding cells, we aimed to synchronize the cells using serum starvation for 24 hr (*Figure 3D*). This approach induces a cell cycle arrest, halting cells in the G0/G1 phase, thereby synchronizing the whole population (*Chen et al., 2012*). We validated the cell cycle arrest by comparing the cell counts of starved conditions with unstarved conditions, which in theory should differ a factor of 2, knowing the cells divide approximately every 24 hr. Indeed, only half of the cell numbers could be detected after 24-hr serum starvation, compared to the corresponding control samples (*Figure 3E*). Interestingly, the percentage of first responding cells obtained from the starved conditions did not significantly differ from the percentages obtained from the unstarved conditions, suggesting that there is no significant effect of cell cycle state on first responders (*Figure 3F*). Additionally, the background levels of IRF7 were (statistically) significantly lower for the starved conditions, compared to the unstarved conditions, again validating our successful approach of starving the cells, which limits the overall protein synthesis (*Figure 3G*).

In short, the extrinsic and intrinsic factors that were assessed in this study turned out to be only minorly dictating the cellular decision to become a first responder. Of note, these results do not exclude other (extrinsic or intrinsic) factors (e.g., those involved in the phosphorylation and translocation of IRF7), those that were not included in this study, from playing important roles in dictating first responders.

## Epigenetic regulation dictating first responders

Our results thus far indicated that stochastic features are only minorly driving first responders, which made us further explore the influence of deterministic features instead. Remarkably, throughout the experiments we observed the occurrence of two neighboring cells showing translocation (*Figure 4A, B*; *Figure 4—figure supplement 1*). If being a first responder is stochastically regulated, the probability of one of the neighboring cells also being a responder is remarkably small, knowing the response rate is only 2.134%. In fact, assuming a cell has on average 4 neighboring cells, the probability of at least one of them being a responder equals the probability of $1 - none\ responds = 1 - 0.97866^4 = 0.0826 = 8.26\%$. Therefore, the observation of responding neighboring cells further supported the hypothesis that first responders are dictated by deterministic, perhaps heritable cell fates. In other words, it seemed more likely that cells that were predispositioned to become a first responder passed this on to their daughter cells, that upon activation both show translocation. Also, after realizing that in the general experimental setup cells were seeded approximately 24 hr before imaging, allowing all cells to have divided once by the time of imaging, the appearance of responding neighboring cells could be further explained and quantified. Accordingly, we can assume that two

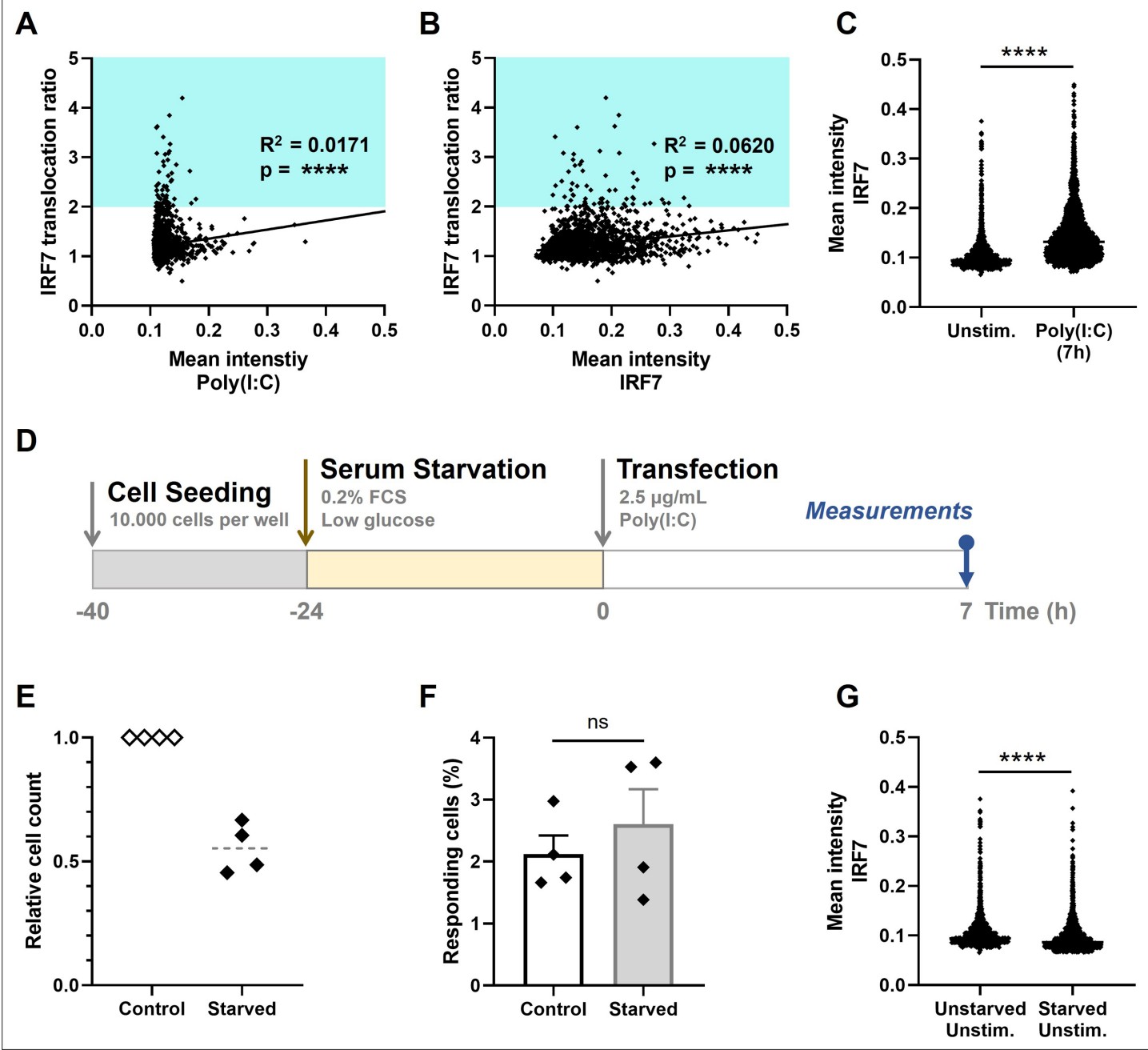

**Figure 3.** Extrinsic and intrinsic stochasticity dictating early type I interferon (IFN-I) responses. (**A**) NIH3T3: IRF7-CFP cells were seeded and transfected as described before. At 7-hr post transfection, images were analyzed using an automated image analysis script to measure rhodamine-labeled Poly(I:C) intensities, and the IRF7 translocation ratios. Plotted are the mean intensities of Poly(I:C) against the IRF7 translocation ratios ($R^2 = 0.0171$). Cyan box indicates IRF7 translocation ratio range accounting for responders. (**B**) As in panel (**A**), the mean intensities of total IRF7 were measured. Plotted are the background levels of IRF7 against the IRF7 translocation ratios ($R^2 = 0.0620$). (**C**) Scatter plot depicting the IRF7 levels of unstimulated cells versus Poly(I:C)-stimulated cells after 7 hr ($p < 0.0001$). (**D**) Experimental design of serum starvation experiments in NIH3T3: IRF7-CFP cells. Cells were seeded 40 hr prior to the start of the experiment. 24 hr prior to transfection, cells were serum and glucose deprived. Next, cells were transfected with 2.5 µg/ ml Poly(I:C) and assessed for nuclear translocation of IRF7 after 7 hr. (**E**) Validation of cell cycle arrest induced by serum starvation by relative cell counts of the control (unstarved) conditions, compared to the corresponding starved conditions ($n = 4$). (**F**) Comparison of the percentages of responding cells of the control conditions, compared to the starved conditions (nonsignificant = ns); data are represented as mean ± standard error of the mean (SEM). ****$p \leq 0.0001$ (Mann–Whitney test). (**G**) Scatter plot of a representative biological replicate comparing the IRF7 levels of unstimulated cells, as in (**C**), versus starved, unstimulated conditions ($p < 0.0001$; Mann–Whitney test, two-tailed).

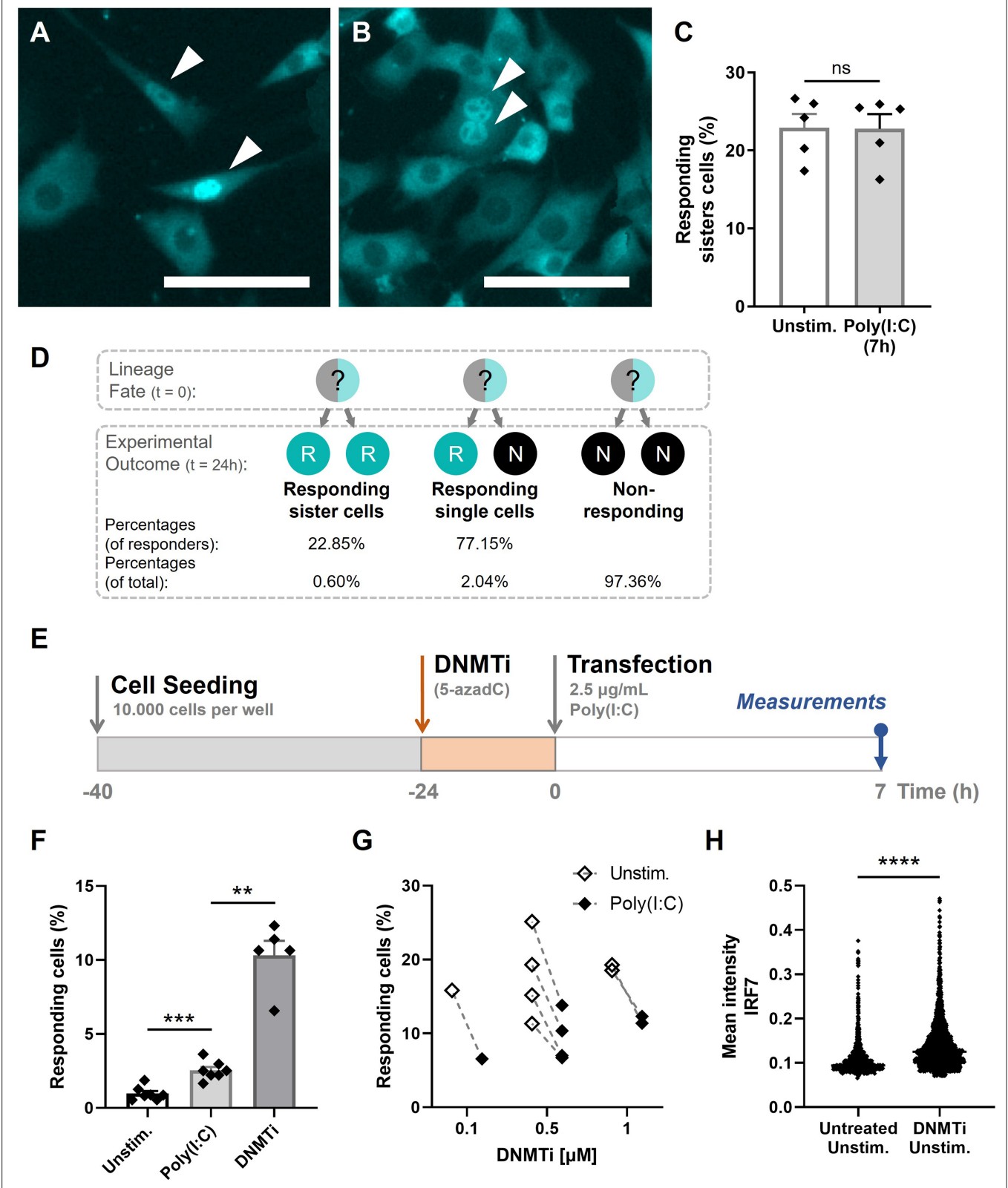

**Figure 4.** Epigenetic regulation dictating early type I interferon (IFN-I) responses. (**A**) NIH3T3: IRF7-CFP cells were seeded on coverslips and transfected with 2.5 µg/ml Poly(I:C) for 7 hr. Microscopy image of two responding, neighboring cells, referred to as responding sister cells, displaying different background levels of IRF7. Scale bar equals 100 µm. (**B**) Microscopy image of two responding sister cells, displaying similar background levels of IRF7. Scale bar equals 100 µm. (**C**) Data on percentages of responding sister cells for unstimulated conditions (background translocation) versus stimulated

*Figure 4 continued on next page*

*Figure 4 continued*

conditions, transfected with Poly(I:C) after 7 hr; data are represented as mean ± standard error of the mean (SEM). **p ≤ 0.01, ***p ≤ 0.001, ****p ≤ 0.0001 (Mann–Whitney test). (**D**) Schematic of theoretical lineage fates and subsequent experimental outcomes (depicted as percentages of responders and of total population) upon cellular division. (**E**) Experimental design of epigenetics experiments in NIH3T3: IRF7-CFP cells. Cells were seeded 40 hr prior to the start of the experiment. 24-hr post transfection, cells were treated with DNMTi to induce hypomethylation. Next, cells were transfected with 2.5 µg/ml Poly(I:C) and assessed for nuclear translocation of IRF7 after 7 hr. (**F**) Percentages of responding cells for unstimulated, stimulated (Poly(I:C)), and DNMTi (1 µM) treated + stimulated conditions; data are represented as mean ± SEM. **p ≤ 0.01, ***p ≤ 0.001, ****p ≤ 0.0001 (Mann–Whitney test). (**G**) Data on paired percentages of responding cells (unstimulated versus stimulated) for different concentrations of DNMTi. (**H**) Scatter plot of a representative biological replicate comparing the IRF7 mean intensity of individual cells of untreated, unstimulated conditions, versus DNMTi treated, unstimulated conditions. **p ≤ 0.01, ***p ≤ 0.001, ****p ≤ 0.0001.

The online version of this article includes the following figure supplement(s) for figure 4:

**Figure supplement 1.** Microscopy images of numerous neighboring cells showing translocation.

**Figure supplement 2.** Percentages of responding cells upon treatment with varying concentration of HDACi TSA, for varying durations.

responding neighboring cells are sister cells, although our current methodology lacks the ability to fully prove that. Interestingly, comparing the two sister cells, the background levels of IRF7 only occasionally differed drastically (*Figure 4A*), whereas for the majority of sister cells the translocation ratios were remarkably similar (*Figure 4B*). Accordingly, a similar phenomenon on a greater intra- than inter-lineage transcriptional similarity has been observed for CD8+ lymphocyte differentiation, which is also considered to be a deterministic process (*Kimmerling et al., 2016*).

Next, we quantified the percentage of responding sister cells (neighboring cells) for both the unstimulated (observed background translocation levels) and stimulated conditions, which were not significantly different from one another (*Figure 4C*). The criteria for responders being assigned as responding sister cells included a maximum distance between the two cells of 300 µm, and a maximum of one nonresponding cells between the two responders. In theory, with an average of 22.85% of responding sister cells, it implies that two responding sister cells originated from one mother cell in 22.85% of the cases (*Figure 4D*). In 77.15% of the cases, only one of the two sister cells turned out to become a first responder. For this scenario, it is yet unclear whether the potential transfer of responder fate (assuming the mother cell was a responder) was only succeeded for only one daughter cell, or whether this single responding daughter appeared stochastically from a nonresponding lineage (assuming the mother cell was a nonresponder). Both have been described in the literature, referred to as transiently heritable cell fates (*Shaffer et al., 2020*).

Continuing the hypotheses of transiently heritable cell fates stated in literature, we investigated the manipulation of cellular decision-making by altering the cells' epigenetic profile, thereby altering any potential predispositioning toward becoming a first responder. Both methylation and histone acetylation have been suggested in dictating transient heritable cellular fates (*Clark et al., 2021*; *Lu et al., 2021*; *Shaffer et al., 2020*). Accordingly, cells were first incubated with DNA methyltransferase inhibitor (DNMTi) 5-Aza-2'-deoxycytidine (5-azadC) 24 hr prior transfection (*Figure 4E*). As indicated in the experimental schematic, the drugs were only administered pre-transfection. We hypothesized that, under regular circumstances, in only 1–3% of cells the epigenetic profiling allows the cell to become a first responder, which is in accordance with similar observations described in literature (*Shaffer et al., 2020*). Accordingly, hypomethylating the DNA of all cells is hypothesized to result in higher response rates. Indeed, cells treated with DNMTis showed higher percentages of first responding cells, arguing that the cellular decision to become a responder is, at least partly, epigenetically regulated (*Figure 4F*; *Figure 4—figure supplement 2*). Also the treatment with varying dosages and durations of Trichostatin A, an histone deacetylase inhibitor (HDACi), increased the number of responding cells. However, unstimulated cells treated with epigenetic drugs also showed increased percentages of responding cells, with even higher percentages upon DNMTi treatment compared to the stimulated DNMTi-treated cells (*Figure 4G*). This might be explained by the effect of DNMTis triggering cytosolic sensing of double stranded RNA originating from retroviruses, that are no longer silenced while using these types of drugs (*Chiappinelli et al., 2015*). We later confirmed this by showing increased levels of IRF7 mean intensities in unstimulated, DNMTi-treated cells, compared to unstimulated untreated cells (*Figure 4H*). Namely, this implies that, though these cells were not transfected with Poly(I:C), these cells got properly activated by the retroviruses, leading to the subsequent production of IFNβ, thereby initiating the positive feedback loops causing higher IRF7 expression levels.

Taken together, we show that, at least partly, the cellular decision-making to become a first responder is epigenetically regulated via both methylation and histone acetylation. Although the self-activation by retroviruses upon hypomethylation might be considered as an artifact, the results still indicate that upon hypomethylation and activation (i.e., either by only retroviruses or in combination with Poly(I:C)), the fraction of first responders increases.

## Fluctuation analysis on first responders

Another elegant approach to assess whether epigenetic mechanisms are involved in driving first responders involves the classical Luria–Delbrück fluctuation test (*Luria and Delbrück, 1943*). It was originally used to demonstrate the occurrence of genetic mutations in bacteria in the absence of selection, rather than being a response to selection, in which variability between different clonal populations is assessed. Similarly, a stochastic feature would be equally present among different clones, whereas a (transiently) heritable feature can widely fluctuate between different clones, depending on the cell fate of the mother cells.

Assuming first responders are purely stochastically regulated, probability calculations can predict from which generation number the probability of at least one first responder present is close to one, knowing that on average only 2.134% first responders are present in a population (see Materials and methods). From generation 6 onwards, the probability of at least one responder being present becomes considerably high. Subsequently, each clone, consisting of ~64 cells, would have 1.37 responding cells on average. To generate the clones of generation 1 through 6, and up, we used low cell seeding (for generation 1–9) and conventional limited dilution approaches (for generations 13 and 16) (*Figure 5A*; *Figure 5—figure supplement 1A, B*). For example, clones of generation 6 were seeded 6 days prior to transfection and imaging, allowing the single cells to have divided six times (generation 6). Of note, in this experimental setting, the generation number is only an indication of the number of cellular divisions that the clone has undertaken, rather than a determinantal factor, as cells do not remain synchronized over multiple generations. Performing a limited dilution for the early clones was practically too challenging. Instead, upon low cell seeding, there is still enough empty space surrounding the clusters of cells to determine which cells originated from a single cell.

Next, clones were stimulated and checked for first responders as described before. Interestingly, the early generation clones showed remarkably high fractions of responders. In fact, the majority (14/18) of generation 1 clones (consisting of two cells) showed 100% responsiveness (*Figure 5B*). With increasing generation numbers, corresponding with increasing cell numbers, the percentage of responders dropped drastically (*Figure 5C, D*; *Figure 5—figure supplement 2*; *Figure 5—figure supplement 3A–D*; *Figure 5—figure supplement 4A–D*). From generation 13 onwards, the percentage of responders did no longer differ significantly compared to the regular clones (generation ∞). These results are similar to the ones described earlier on subpopulations of cancer cells purified for a given phenotypic state to return toward equilibrium proportions over time (*Gupta et al., 2011*). In our experiments, up till clones of generation 6, the fluctuation across clones was rather large, with some clones showing no single translocation event. The rather long timescales of switching from responders to nonresponders, and the other way around, imply epigenetic mechanisms at play, and indeed, prior work has indicated an important role for epigenetics dictating IFN-I response dynamics (reviewed in *Barrat et al., 2019*).

Altogether, the results from the fluctuation assay indicate a yet unknown, deterministic phenomenon which dictates responsiveness, which seems to overrule stochasticity.

## Modeling first responder cellular decision-making across generations

For a proper interpretation of the results obtained from the fluctuation assay, we modeled cellular decision-making during early IFN-I responses, where individual cells are either displaying IRF7 translocation, making them first responders, or not. Assuming a purely stochastic process, upon cloning, the total mean across clones should be equal to the mean obtained from regular cultures, which would be 2.134% (*Figure 6A*). Accordingly, the coefficient of variation (CV) is determined by the biological and technological variation, therefore considered relatively low. The rate in which responders appear in the population ($k_{on}$) is also relatively low, corresponding with the probability of a cell to become a responder (p = 0.02134). Assuming a strictly heritable fate, meaning that all responding cells will divide into responding daughter cells, the total mean across clones will not change (*Figure 6B*).

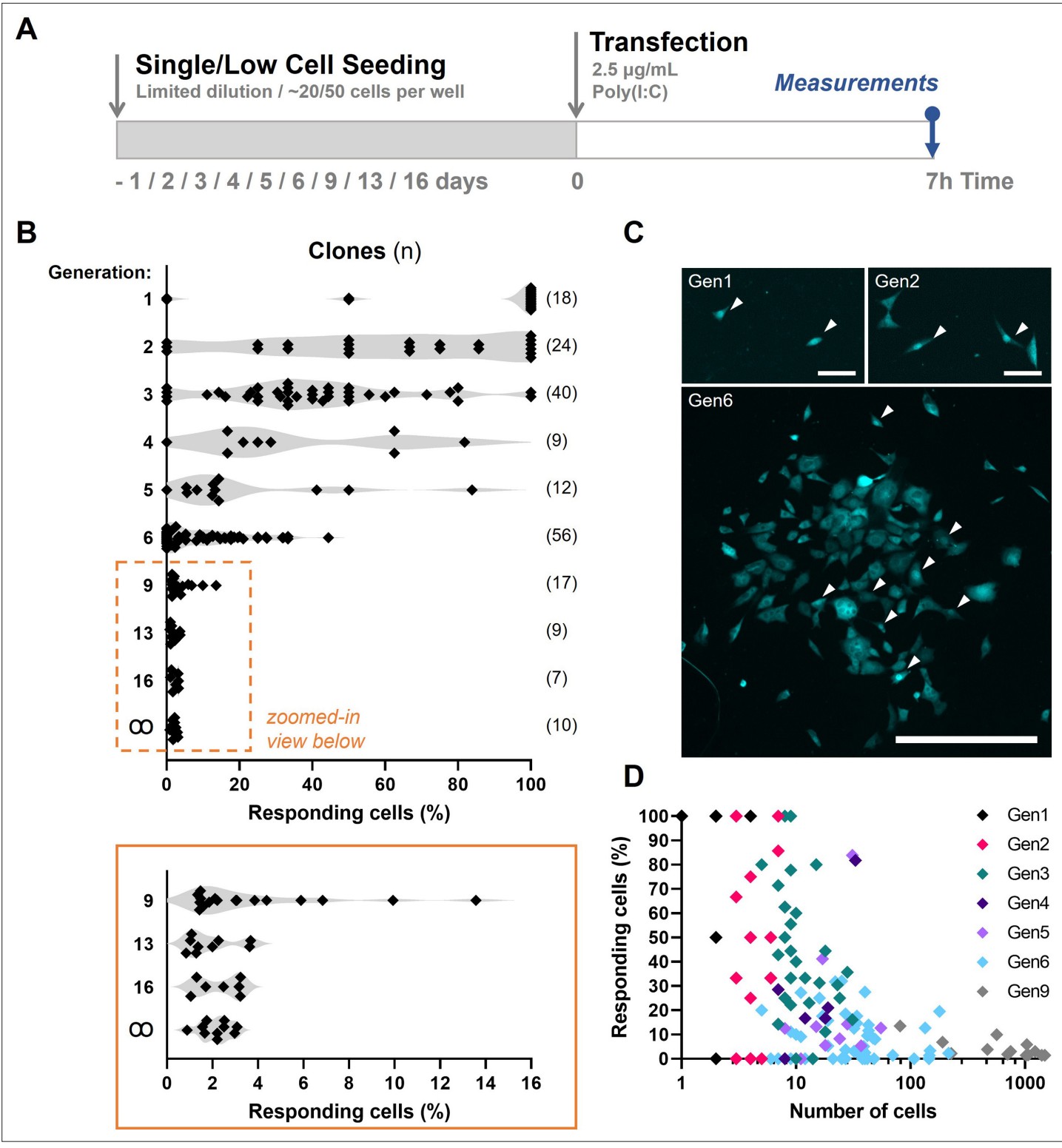

**Figure 5.** Fluctuation analysis on first responders. (**A**) Experimental design of fluctuation experiments in NIH3T3: IRF7-CFP cells. Cells were either seeded following limited dilution or at only ~50 cells per 24-well, depending on the generation number. Next, cells were transfected with 2.5 µg/ml Poly(I:C) and assessed for nuclear translocation of IRF7 after 7 hr. (**B**) Fluctuation plots on percentages of responding cell of clones of different generations. Generation ∞ equal regular cultures. (**C**) Microscopy images of clone of generations 1 (gen1), 2 (gen2), and 6 (gen6) displaying varying percentages of translocated cells, some of which are indicated with white arrows. (**D**) Scatter plot on data obtained from clones of generation 1 through 9.

*Figure 5 continued on next page*

*Figure 5 continued*

The online version of this article includes the following figure supplement(s) for figure 5:

**Figure supplement 1.** Probability calculations.

**Figure supplement 2.** Statistical analysis on fluctuation data.

**Figure supplement 3.** Microscopy images of clones of generation 6.

**Figure supplement 4.** Microscopy images of clones of generation 9.

However, the CV will be much higher than the biological and technological noise, determined by the occurrence of responding lineages. The $k_{on}$ is not defined, as individual cells will no longer change fate across the generations.

While a purely stochastic cellular decision and a strictly heritable cellular decision cover two extremes, the phenomenon of transiently heritable cell fates is characterized by a type of heritability that falls between those two ends of the spectrum (*Lu et al., 2021*; *Shaffer et al., 2020*). In fact, transiently heritable cell fates cover an intermediate timescale, in which cellular states may persist for several cellular divisions but are ultimately transient, and thus not indefinitely heritable. Still, this phenomenon can clearly be distinguished from the rather short-lived fluctuations referred to as noise (*Shaffer et al., 2020*). As a transiently heritable phenomenon allows responders to appear from nonresponding parental cells, the mean across clones will still equal to 2.134%, while the CV will be relatively high too, but not as high as compared to a strictly heritable fate (*Figure 6C*). The $k_{on}$ will be based on the probability of the reintroduction of responding cells, which can be variable, but should per definition be slower than the rate of cell division.

Surprisingly, the data obtained from clones of generation 1 through 9 resulted in a mean higher than 2.134% (*Figure 6D*), and a fluctuating CV (*Figure 6E*). From generation 13 onwards, both the mean and the CV start to meet the data obtained from the regular cultures again, which are similar to the theoretical outcomes of a stochastic process. Accordingly, we modeled first responders as a binary switch, where individual cells are either responding (ON) or nonresponding (OFF), similar to the transient heritable fates characterized and modeled before (*Shaffer et al., 2020*). Details on the ODE model are provided in the Materials and methods. We could fit the transient heritability model to the data when starting from 100% responders at generation 0 (i.e., a single cell isolated from the regular culture). Cells switch OFF after five generation on average, with a constant $k_{on}$ rate throughout. Interestingly, in generation 0 we observed (nearly) only IFN-I responders, which we believe might be caused by single cells being deprived from any paracrine cues, which could include inhibitory factors that normally limited responsiveness. However, single IFN-I-producing cells (i.e., plasmacytoid DCs and monocyte-derived DCs) encapsulated in picoliter droplets or captured in small microfluidic chambers did not display this behavior (*Shalek et al., 2014*; *Wimmers et al., 2018*). Instead, one could argue that single cells establish a different microenvironment, compared to a situation in which cells are close to neighboring cells, which elicits behavioral changes accordingly. The dimensions of microfluidic droplets and chambers are in the same range of cell-to-cell contacts in vitro, while single cells seeded for cloning are surrounded by rather massive areas and volumes without other cells present. Therefore, we hypothesize that these single cells lack biochemical, and perhaps biomechanical cues provided by dense cell populations, which result in behavioral changes in these cells, in our case, making them more responsive. Similarly, in quorum sensing, cells secrete soluble signaling molecules (called autoinducers), which enables cells to get a sense of their cell density (*Postat and Bousso, 2019*; *Waters and Bassler, 2005*). Without signaling of these molecules, cells perceive being isolated from the rest. In microfluidic droplets and chambers, these molecules accumulate, given the relatively small volumes.

Together, we validated transiently heritable cellular decision-making driving responders using mathematical modeling.

## Quorum sensing drives cellular decision-making during early IFN-I responses

One possible explanation for the observed higher responsiveness in early generation clones, like described before, lies in the possible effect of cell density dictating responsiveness, which closely relates to the phenomenon called quorum sensing. Intuitively, the immune strategy in which the

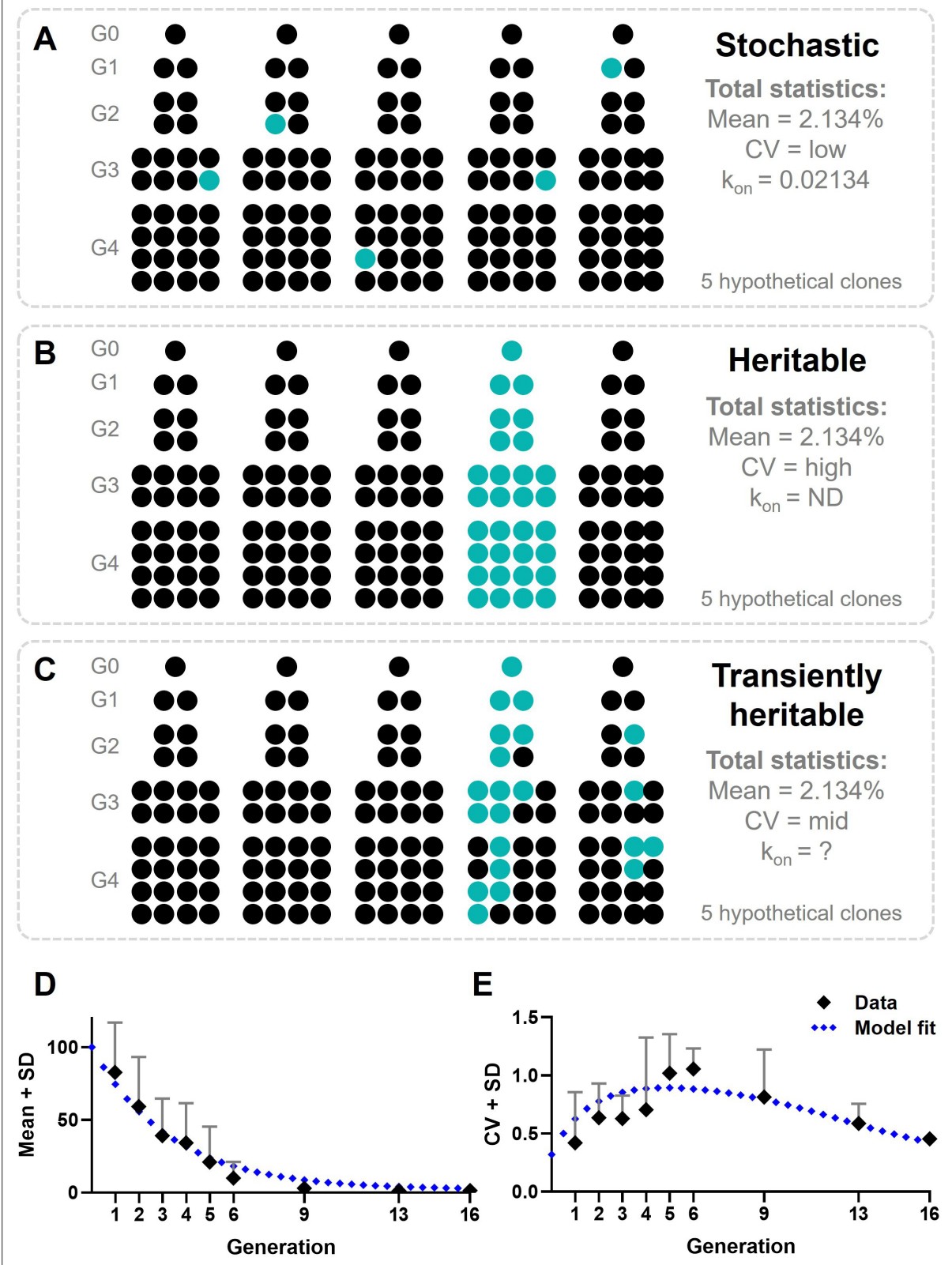

**Figure 6.** Modeling cellular decision-making during early type I interferon (IFN-I) responses. (**A**) Hypothetical outcomes of responding cells upon cloning, assuming cellular decision-making during early IFN-I responses is a stochastic process. Responders will appear randomly across clones, resulting in a total mean of 2.134%, a low coefficient of variation (CV), and a low $k_{on}$. (**B**) Hypothetical outcomes of responding cells upon cloning, assuming cellular decision-making during early IFN-I responses is a heritable process. Responders will appear only from the lineage that started with a

*Figure 6 continued on next page*

*Figure 6 continued*

responding cell, resulting in all offspring becoming responders. The total mean will still be 2.134%, though the CV will be high and the $k_{on}$ will be zero, as no cell switch fate. (**C**) Hypothetical outcomes of responding cells upon cloning, assuming cellular decision-making during early IFN-I responses is a transiently heritable process. Responders are more likely to appear in lineages originating from a responding cells, but can also appear in lineages that started with a nonresponding cell. Besides, responding cells can also disappear from responding lineages. This results in a total mean of responding cells that higher than 2.134%, with a high CV, and a variable $k_{on}$. (**D**) Mean plus standard deviation (SD) of experimental outcomes of fluctuation assay with ordinary differential equation (ODE) model fitted; data are presented as mean ± SD. (**E**) CVs of fluctuation assay with ODE model fitted.

fraction of responders, in this case IFN-I producers, is based on the amount of available cells seems crucial to establish proper antiviral immunity at any circumstance (*Van Eyndhoven and Tel, 2022*). Accordingly, a small population of cells needs to contribute to greater extend, involving relatively more responders, than a large population of cells, to ensure the overall IFN-I production is similar. In our experiments, ranging from clones of generation 1 toward generation 13, the cell density (absolute cell count per area/volume) increases exponentially. Therefore, we hypothesized that at a lower cell density, corresponding with low generation numbers, cells tend to be programmed to become more responsive, meaning that percentages of responding cells become higher.

The phenomenon of different cellular behaviors upon differences in cell density is in agreement with the concept of (immune) quorum sensing, which describes the ability of (immune) cells to perceive the density of their own population and adjust their behavior accordingly (*Antonioli et al., 2019*; *Polonsky et al., 2018*). Subsequent alterations in responsiveness are thought to be coordinated via epigenetic regulations. As we previously indicated a role for epigenetics driving first responders, we wondered whether we could explore the effects of quorum sensing in cellular decision-making during early IFN-I responses. Therefore, we hypothesized that cellular decision-making is defined by epigenetic profiling, which allows switching over time between a responding and nonresponding state, even before stimulation, and is subject to the phenomenon of quorum sensing.

To test this final part of our hypothesis, we generated clones of generation 6 in low and high densities on coverslips as described before (*Figure 7A*). We hypothesized that clones at low seeding densities display more fluctuations in the percentage of responders compared to high seeding densities, based on the results obtained in the fluctuation assay. Low seeding densities were obtained by seeding 250 cells per 24-well and verified upon visual inspection, meaning that these clusters of cells did not exceed the expected cell count of single clones ($2^{6/7}$ = 64/128 cells, depending on their grow speed), and were clearly separated from other clusters of cells, with over a 1400-μm distance between the center points of the clones (*Figure 7B, C*; *Figure 7—figure supplement 1A–C*). High seeding densities were obtained by seeding 1000 cells per 24-well, which resulted in merged groups of clones, thereby evidently exceeding the expected cell counts per cluster (*Figure 7D, E*). In practice, clones seeded at high sending densities occasionally led to single clones, as observed upon low cell seeding. For these instances, these clusters were considered as a single clones.

The results confirmed that single clones of generation 6 displayed high fluctuation, which closely matched with the data obtained earlier (average of 10.67 compared to 10.81; CV of 0.87 compared to 1.04, respectively) (*Figure 7F*). Interestingly, the averages of merged clones of generation 6 displayed percentages of responding cells which closely matched with the numbers obtained from regular cultures (1.96% compared to 2.13, respectively).

To conclude, we confirm that cellular decision-making during early IFN-I responses is likely affected by the effects of quorum sensing. In other words, cell seems to be aware of their density, and adjust their epigenetic profiling to allow their secretory behaviors accordingly (*Figure 7G*).

## Discussion

Here, we assessed the role of host-intrinsic factors dictating early IFN-I response dynamics. We observed that the cellular decision to become a first responder can be considered as a fate, likely driven by a distinct epigenetic profile, rather than a coincidence driven by stochastic factors. Furthermore, this fate seems transiently heritable, of which the timescale indicates epigenetic mechanisms at play, which is in line with the results obtained using epigenetic drugs. Previously, the overall consensus on how first responders were thought to be regulated was by stochastic regulation, or in other words randomly, which is now challenged (*Wimmers et al., 2018*). This rare fraction of cells has been

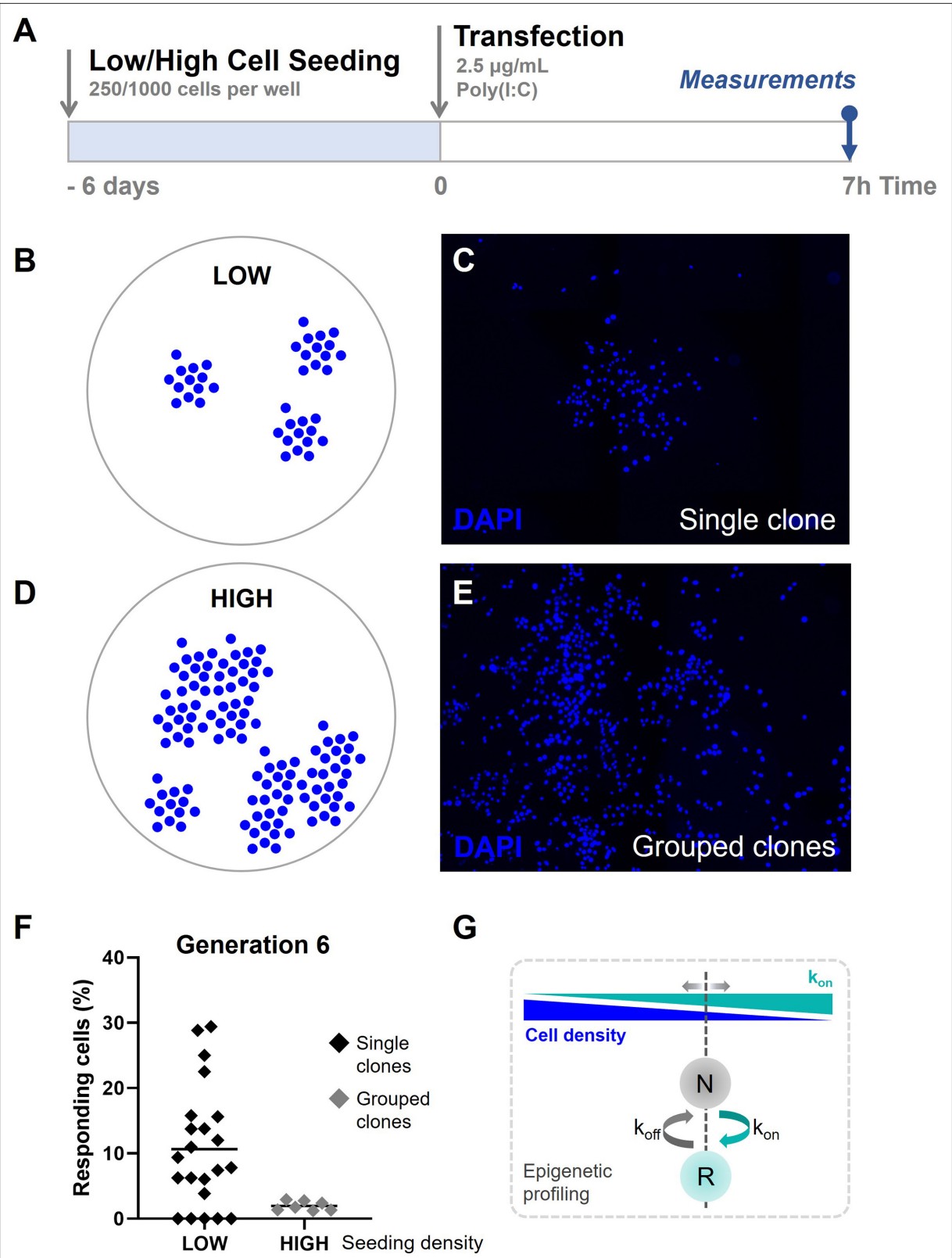

**Figure 7.** Quorum sensing drives cellular decision-making during early type I interferon (IFN-I) responses. (**A**) Experimental design of quorum sensing experiments with NIH3T3: IRF7-CFP cells. Cells were either seeded at low numbers or high numbers (250 versus 1000 cells per 24-well) 6 days prior to the start of the experiment. Next, cells were transfected with 2.5 µg/ml Poly(I:C) and assessed for nuclear translocation of IRF7 after 7 hr. (**B**) Schematic representation of single clones of generation 6 on coverslips, seeded at low cell densities. (**C**) Microscopy image of DAPI (4',6-diamidino-2-phenylindole)

*Figure 7 continued on next page*

*Figure 7 continued*

channel, visualizing the nuclei of cells, displaying clear clustering of single clones of generation 6. (**D**) Schematic representation of grouped clones of generation 6 on coverslips, seeded at high cell densities. (**E**) Microscopy image of DAPI channel, visualizing the nuclei of cells, displaying grouped clusters of cells, consisting of numerous clones of generation 6. (**F**) Scatter plots on percentages of responding cell of clones of generation 6 seeded in low densities (*n* = 22 clones), and grouped clones seeded at high densities (*n* = 7). (**G**) Schematic on cell fate switching, influenced by cell density.

The online version of this article includes the following figure supplement(s) for figure 7:

**Figure supplement 1.** Microscopy images of cells of generation 6 seeded at varying seeding densities.

described as indistinguishable from the rest, except in their expression of core antiviral gene expression programs (*Shaffer et al., 2020*; *Shalek et al., 2014*). In contrast, we currently hypothesize that first responders are predetermined, meaning that their epigenetic profile is driving their responder fate.

Not only the epigenetic profile drives responsiveness, but also the effect of cell density, often referred to as quorum sensing. Because only a fraction of the cells become a first responder, quorum licensing might be a more suitable word of choice, as typically quorum sensing refers to a digital outcome in which either all cells or none at all respond (*Muldoon et al., 2020*). The exact contribution of heritability versus quorum sensing/licensing driving responsiveness can be further dissected using single-cell epigenetic profiling, in combination with lineage tracking, in density-controlled settings.

The phenomenon of a small fraction of first responders, responsible for the rapid and robust production of IFN-Is, has been observed across species and cell types. Moreover, various stimuli (live and synthetic) targeted membrane, cytosolic, and endosomal receptors, arguing that the mode of activation is not driving the discrepancies in responder fates. Therefore, we consider our utilized murine cell model as a good immune-cell or generic tissue-cell alternative for characterizing the fundamentals of cellular decision-making upon viral infection. Future studies have to translate the fundamental findings obtained in this study toward in vivo applications, and potentially, toward clinical applications. However, we do speculate that, in vivo, quorum sensing triggers collective and coordinated actions, as observed in DCs activated in the draining lymph node, in vivo (*Bardou et al., 2021*). Intuitively, it makes sense that cell density plays a major role in cytokine-mediated responses, while the same antiviral effects are achieved by a high responding fraction of a small population of potential producers, as compared to a low fraction of a large population of potential producers (*Van Eyndhoven and Tel, 2022*). Although the translation of single-cell work might seem challenging, because of the seemingly unnatural situations mimicked with single-cell work, we believe that the fundamentals of cellular decision-making are similar across numbers, scales, and systems. Accordingly, studies in small intestinal organoids report similar bimodal IFN-I responsiveness (*Bhushal et al., 2017*). Likewise, transcription factor nuclear factor kappa B (NF-κB) translocation follows similar all-or-nothing (i.e., digital) response dynamics, is prone to epigenetic licensing, and corresponding responding fates have proven to be transiently heritable as well (*Clark et al., 2021*).

Cells are faced by many decisions in response to external stimuli, reflected by a massive degree of cellular heterogeneity. By sharing information, a population of cells can make more effective decisions compared to each individual cell alone (*Perkins and Swain, 2009*). The ability of a fraction of first responders to drive population-wide IFN-I dynamics via paracrine signaling may be an efficient and robust strategy for quorum sensing, which allows tight regulation, but at the same time allows for flexibility and adjustability (*Shalek et al., 2014*). At the same time, this immune strategy is prone for mistakes. In autoimmune diseases like systemic lupus erythematosus, excessive IFNβ production potentiates auto-reactive DC activation (*Hall and Rosen, 2010*; *Muskardin and Niewold, 2018*). In contrast, excessively stringent thresholds may limit rapid responses to viral infection, as observed during severe acute respiratory syndrome coronavirus 2 (SARS-CoV-2) infection (*Park and Iwasaki, 2020*; *Ziegler et al., 2021*). Therefore, cellular decision-making regarding the production of these reactive, potential harmful cytokines is inherently complex.

The additional mechanisms underlying the transiently heritable fate to become a first responder remain yet mysterious. Further work will be required to assess potential modules of gene regulation dictating these rare cellular decision-making processes, such as methylation or other regulatory mechanisms that operate on intermediate timescales (*Meir et al., 2020*). Perhaps, the transient heritability of epigenetic profiles is rather small, while stochastic protein states are also transmitted from mother to daughter cells, driving responder fates as well (*Spencer et al., 2009*). Additionally, our utilized

methodologies were only semi-quantitative. Perhaps, single molecule fluorescence in situ hybridization could provide an enhanced resolution to the cellular decision-making observed in this study. Regarding the phenomenon of quorum sensing, studies have reported that DC activation by Poly(I:C) harbors a collective production of IFN-I, which drives DC activation at the population level in vivo (*Bardou et al., 2021*). Therefore, sustained IFN-I signaling in mediating full DC activation promotes collective behaviors, instead of cell-autonomous activity. In other words, the concentration of IFN-Is produced by a single cell has no biological effect, whereas the accumulation of IFN-Is produced by many cells drive a collective response. This makes us hypothesize that a low cell density, as occurs for clones of generation 6 in the fluctuation assay, increases the percentage of first responder fates, thereby avoiding the risk of IFN-I levels that are too low to have any biological effect.

While transcriptional regulators have been the main focus of studying IFN-I dynamics, insights on additional types of regulation, such as epigenetic regulators and quorum sensing are shedding their light on an already complex IFN-I system. Although the presence of first responders mainly got characterized using microfluidic techniques, which seem far from representing the complex in vivo situation, studies have proven their existence and importance in vivo (*Bauer et al., 2016*; *Zhang et al., 2020*). Additionally, understanding the fundamentals of cellular decision-making during early IFN-I responses open compelling avenues for future development of novel IFN-I-targeted therapies. Especially considering the crucial role of well-orchestrated IFN-I response dynamics in clearing SARS-CoV-2 infection, while preventing harmful and ineffective cytokine storms (*Park and Iwasaki, 2020*), emphasizes the necessity of understanding the fundamentals of cellular decision-making. Together, the combination of single-cell technologies, mathematical modeling approaches, and the in vivo validation and translation continues to unravel the complexity of the IFN-I system in physiological contexts.

# Materials and methods

**Key resources table**

| Reagent type (species) or resource | Designation | Source or reference | Identifiers | Additional information |
|---|---|---|---|---|
| Cell line (*Mus musculus*) | NIH3T3: IRF7-CFP | Obtained from Ulfert Rand and Mario Köster, Helmholtz Centre for Infection Research, Germany | | *Rand et al., 2012* |
| Cell line (*Mus musculus*) | NIH3T3: STAT1-CFP, STAT2-YFP | Obtained from Ulfert Rand and Mario Köster, Helmholtz Centre for Infection Research, Germany | | *Rand et al., 2012* |
| Chemical compound, drug | Lipofectamine2000 | Invitrogen | cat #12566014 | |
| Chemical compound, drug | Poly(I:C) (LMW) Rhodamine | InvivoGen | cat #tlrl-piwr | |
| Chemical compound, drug | 5-Aza-2′-deoxycytidine | Sigma-Aldrich | cat #A3656 | |
| Chemical compound, drug | Trichostatin A | Sigma-Aldrich | cat #T8552 | |
| Software, algorithm | CellProfiler | https://www.cellprofiler.org | | *Stirling et al., 2021* |

## Cell culture and activation

Reporter murine fibroblastoid NIH 3T3 cells with stable expression of IRF7-CFP, STAT1-CFP, and STAT2-YFP fusion proteins were provided by Ulfert Rand and Mario Köster (Helmholtz Centre for Infection Research, Germany). Both cell lines have been authenticated and checked for mycoplasma contamination. Cells were cultured under standard tissue culture conditions in Dulbecco's Modified Eagle Medium (DMEM; Sigma) supplemented with 10% fetal calf serum, glutamine, penicillin, streptomycin, and selection antibiotic G418 or puromycin. pIRF7-CFP, pSTAT1-CFP, and STAT2-YFP were created by introducing cDNA (C57BL/6) in a pMBC-1 vector containing linker and sequences for CFP and YFP via EcoRI restriction sites (*Dirks et al., 1994*). Transfections of plasmid DNA were performed with Metafectene (Biontex) according to the manufacturer's instructions. G418- or puromycin-selected representative clonal cells showing stable expression of the reporter construct and strong signal to background ratio of the fluorescent marker were used. For experiments, cells were seeded on glass

coverslips in 24-well plates, and activated using Lipofectamine2000 (Invitrogen) transfection reagent according to the manufacturer's instructions. At all times, fluorescently labeled stimuli (rhodamine-labeled LMW Poly(I:C), InvivoGen) were used to assess transfection timing and efficiencies throughout the experiments. For additional transfection optimization, cells were analyzed using confocal microscopy (Nikon Eclipse Ti2), and measured with a flow cytometer (FACS Canto).

## Image and data analysis

Coverslips with cells were thoroughly washed (3×) with medium containing 10% fetal calf serum to loosen sticky liposomes from the glass and from the cell's surfaces, to avoid false positivity upon assessing transfection efficiency. Next coverslips were fixed with 3% formaldehyde for 15 min at room temperature, washed, and stained with Hoechst 33343 to visualize nuclei. Next, coverslips were mounted on microscopy slides using Vectashield mounting media (Vector Laboratories), and imaged with a Nikon Eclipse Ti2 fluorescent microscope (Nikon). Image acquisition was performed by making multi-tile images at a magnification of ×20. Images were analyzed with ImageJ (National Institutes of Health) and a customized CellProfiler script (https://www.cellprofiler.org). Transfection efficiencies were determined based on the mean intensities provided by the CellProfiler script. The transfection threshold was based on the maximum intensity obtained from the untransfected cells. IRF7 translocation ratios were calculated using the following equation:

$$IRF7\ translocation\ ratio = \frac{Nucleus_{CFP\ median\ intensity}}{Cytoplasm_{CFP\ median\ intensity}}$$

Images from which the percentage of translocated cells were drawn were at all times manually and visually checked, considering the relatively low percentages. Besides, the translocation ratio threshold for distinguishing responders from nonresponders was optimized per experiment, again based on visual validation. Data visualization and statistical analysis were performed using the GraphPad Prism software (GraphPad).

## Fluctuation assay

Single cells were seeded into 96-well plates using limited dilution in regular growth medium supplemented with 20% fetal calf serum and 20% conditioned medium obtained from regular cultures. Upon cell stretching, all wells were visually inspected to detect multiple seeded cells per well, and excluded from the experiments. For sixth and ninth generation clones, cells were seeded on glass coverslips in a concentration of 10 or 50 cells per well, respectively, and tracked over time to assure single-cell clones. Probability calculations were performed using the following equations:

$$P\left(1 \leq first\ responders\right) = 1 - P\left(no\ responders\right)$$

$$P\left(no\ responders\right) = fraction_{nonresponders}^{2^{generation}}$$

## Mathematical modeling

We consider a simple model where single cells can be in either one of two states: responsive and unresponsive. Cells in the unresponsive state become responsive with rate $k_{on}$, and responsive cells become unresponsive with rate $k_{off}$. At steady-state, only $f = 2\%$ of cells are in the responsive state implying

$$\frac{k_{on}}{k_{on}+k_{off}} = f \implies k_{on} = k_{off}\frac{f}{1-f}$$

Our data show that at the start of the fluctuation test experiment, single cells are mostly in the responsive state, and as the colony proliferates, the fraction of responsive cells converge back to $f$ over time. The average fraction of responsive cells $x\left(t\right)$ over time is given by the ODE

$$\frac{dx}{dt} = k_{on}\left(1 - x\left(t\right)\right) - k_{off}x\left(t\right)$$

Assuming an initial condition $x\left(0\right) = 100\%$, we fit the solution of this equation

$$x\left(t\right) = f + \left(1 - f\right) e^{-\left(k_{on} + k_{off}\right)t} = f + \left(1 - f\right) e^{-\frac{k_{off}t}{1-f}}$$

to the mean fraction of responsive cells over time to obtain $k_{off} \approx 0.29\, \text{days}^{-1}$ that corresponds to the average time $1/k_{off}$ in the responsive state to be $\approx 3.5\, \text{days}$ with a 95% confidence interval of $(2.8,\, 4.1)$ days (**Figure 6D**). Since $f \ll 1$, for the initial time points

$$x\left(t\right) \approx e^{-k_{off}t} \implies Log\, x\left(t\right) \approx -k_{off}t$$

and we performed a linear regression between $Log\, x\left(t\right)$ and $t$ using the data from days 0 to 5 to get the 95% confidence interval for the slope $k_{off}$.

Having estimated the kinetics of switching, we next considered a stochastic formulation of the model, where the time individual cells stay in the responsive (unresponsive) state is an exponentially distributed random variable with mean $1/k_{off}$ $(1/k_{on})$. We refer the reader to Saint-Antoine et al. for mathematical details on the stochastic model (**Saint-Antoine et al., 2022**). The colony-to-colony fluctuations $CV_{model}$ in the fraction of responsive cells (as quantified by the CV) was obtained by solving equations 12, 13, and 23 in Saint-Antoine et al. assuming that the initial single cell was in the responsive state. To account for the technical noise, we further modify this equation to

$$CV^2 = CV_{model}^2 + CV_{tech}^2$$

where $CV_{tech} \approx 0.32$ is the fluctuations in the fraction responsive cells between independent bulk samples. The model-predicted $CV$ matches the measured inter-colony fluctuations over time (**Figure 6E**). The stochastic tuning off of cells from responsive to unresponsive states results in the $CV$ first increasing with time to reach a maximum at day 5, and then it monotonically decreases to the technical noise levels.

## Acknowledgements

The authors would like to thank Ulfert Rand, Hansjörg Hauser, and Mario Köster for providing the reporter cells. Additionally, the authors would like to thank Nidhi Sinha and Bart M Tiemeijer for the enthusiastic, insightful, and lively discussions. This work was supported by the European Research Council (ERC) under the European Union's Horizon 2020 research and innovation program (grant agreement no. 802791). Finally, the authors would like to acknowledge the generous support by the Eindhoven University of Technology.

## Additional information

### Funding

| Funder | Grant reference number | Author |
| --- | --- | --- |
| Horizon 2020 | 802791 | Jurjen Tel |

The funders had no role in study design, data collection, and interpretation, or the decision to submit the work for publication.

### Author contributions

Laura C Van Eyndhoven, Conceptualization, Formal analysis, Validation, Investigation, Visualization, Methodology, Writing – original draft, Writing – review and editing; Vincent PG Verberne, Formal analysis, Investigation, Methodology; Carlijn VC Bouten, Supervision; Abhyudai Singh, Conceptualization, Validation, Methodology, Writing – original draft; Jurjen Tel, Conceptualization, Supervision, Funding acquisition, Validation

### Author ORCIDs

Laura C Van Eyndhoven ⓘ http://orcid.org/0000-0001-7230-1134
Abhyudai Singh ⓘ http://orcid.org/0000-0002-1451-2838
Jurjen Tel ⓘ http://orcid.org/0000-0002-7213-3422

**Decision letter and Author response**
Decision letter https://doi.org/10.7554/eLife.83055.sa1
Author response https://doi.org/10.7554/eLife.83055.sa2

## Additional files

### Supplementary files
• Transparent reporting form

### Data availability

The raw data supporting the conclusions of this article are available on DataDryad.

The following dataset was generated:

| Author(s) | Year | Dataset title | Dataset URL | Database and Identifier |
|---|---|---|---|---|
| Van Eyndhoven LC | 2023 | Raw Data Total | https://dx.doi.org/10.5061/dryad.2547d7wtz | Dryad Digital Repository, 10.5061/dryad.2547d7wtz |

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
