## [Editor Report]

This important study combines quantitative experiments and modeling to dissect the factors guiding cell fate decisions during early antiviral (type I interferon) signaling. The authors provide solid evidence that the fate of cells is transiently heritable, and uncover a role for cell density in regulating responsiveness, reminiscent of a quorum sensing mechanism. This work will be of broad interest to systems biologists, immunologists, and cell biologists.

---

## [Decision Letter]

**Decision letter after peer review:**

Thank you for submitting your article "Transiently heritable fates and quorum sensing drive early IFN-I response dynamics" for consideration by *eLife*. Your article has been reviewed by 3 peer reviewers, and the evaluation has been overseen by a Reviewing Editor and Carla Rothlin as the Senior Editor. The following individuals involved in review of your submission have agreed to reveal their identity: José Ordovas-Montanes (Reviewer #1); Yogesh Goyal (Reviewer #2); Sydney Shaffer (Reviewer #3).

Essential revisions:

Overall, the Reviewers agreed this is a well-designed, well-controlled, and timely study that presents important findings, which add to a growing body of evidence reporting heritable cell states in guiding cell fate choices. The experiments are well executed and clear, supporting the general claim that IFN-I early responder events are non-stochastic. However, the Reviewers were also in agreement about the limitations of the findings, specifically the generalizability of the finding and the link to epigenetic regulation, and shared concerns that some of the claims and terminology were not supported by the experiments. Finally, the Reviewers all raised important points about enhancing the clarity of the manuscript. The authors are expected to address the following points in their revision:

1) Reviewers 1 and 3 both raised questions on the generalizability of the finding to other cell types, experimental systems, contexts. The authors are expected to address this comment either experimentally or, at the least by, by pointing to other relevant work.

2) Reviewers 1 and 2 both raised important questions on the contribution of epigenetic regulation on first responder fate, specifically the lack of mechanistic links to DNA methylation. Suggestions for experiments are made below to further verify that epigenetic regulation modulates responder frequency.

3) Reviewers 1 and 2 raised relevant points on the limitations of the modeling and fitting which should discussed. Additionally, Reviewer 2 has suggested the authors temper their claims of "quorum sensing" and instead consider revising their language to call their result "density dependent".

4) Throughout their reviews, the Reviewers have made good suggestions for enhancing the clarity of the manuscript, specifically the need to provide clear definitions upfront, clarity in figures and legends, and elaboration on interpretations of their results (e.g. Figure 4G).

*Reviewer #1 (Recommendations for the authors):*

Additional Suggested Experiments

• Reproduce the generational variability and quorum sensing experiments in a distinct cell type or primary cells to confirm that transient heritability is a more general feature of IFN-I response and understand differences by cell lineage.

• Further verify that epigenetic regulation (and not solely retrovirus activation) modulates responder frequency using either (1) an orthogonal epigenetic drug (e.g., histone deacetylase inhibitor) or (2) evaluating the addition of retroviruses into cells.

• We propose the following extensions to help validate and improve the work, but they may be beyond the scope of available technologies that would be readily implemented:

o Lineage tracing experiments using sequenceable barcodes to directly measure mother-daughter relationships and heritability

o Measurement of DNA methylation, histone modification, or chromatin accessibility in first responders vs non-responders with and without DMNTi

*Reviewer #2 (Recommendations for the authors):*

My comments are listed below, in no particular order:

Could the authors more clearly state how the cutoff was defined for what IRF7 translocation ratio for responders versus non-responders? As it stands, it is unclear how the threshold was decided.

Figure 1E please label that X axis as IRF7 translocation ratio. Also, the current method of representation in 1E presents a high cognitive burden.

The authors should consider using simpler visualization to make this point.

Figure 2C legend can you define what the red dotted line and orange line signifies?

Please add scale bars to all figures. Not a single image in the main text seemed to have scale bars.

The quantifications of mean intensities in Figures 3 and 4 are semi-quantitative at best. While I do not expect the authors to address this point experimentally (with a more quantitative method such as smFISH), they should highlight this as a limitation or add it as a discussion point somewhere.

The authors do not strictly identify "sister" or "twin" cells. Rather, they use neighboring cells as a proxy. Unless the authors actually trace sister cells in this study, this terminology should be avoided in the paper as it is misleading and incorrect.

Along the same lines – how clear is it that cells do not move much for their neighboring cell analysis to be an accurate representation of progeny?

The authors have ignored several seminal studies when inspiring the idea of transient cell states guiding fate decisions. The authors are encouraged to read and mention some of these early studies (Spencer et al. 2009, Gupta et al. 2011, Sharma et al. 2010, and on LD: Fiddler and Kripke 1977) and some more recent ones on heritable fluctuations (Oren et al. 2021, Fennell et al. 2022, Quinn et al. 2022).

In the introduction, some of the terms are poorly defined or used without defining. One example is "multilayered stochasticity". Another example is contrasting stochasticity with determinism, which especially in the introduction is a bit misleading. The authors do clarify the nuance in the Results section briefly (lines 215-227) but the Introduction warrants a more nuanced discussion of these deep conceptual ideas. For example, even such heritable fluctuations, in principle, can arise from stochastic interactions (Schuh et al., 2020).

While the results with the DNMTi are quite interesting, a more detailed discussion and rationale of why demethylation was specifically chosen as a method to modulate the percentage of first responders would be helpful to put these results in context.

In the experiments with DNMTi, the authors can clarify if the DNMTi was continuously provided or media was replaced before infection? In general, again, I do not expect the authors to do these experiments necessarily, but it would be good to comment on or hypothesize how their results may look like for pre-treatment, co-treatment (i.e. DNMTi added right when polyIC is added), or both for DNMTi.

Have the authors considered an experiment where they sort the high responders and see if they lose this ability over time? Even if the authors do not perform the experiment, I am curious what the authors think of this design.

It is outside the scope of this study experimentally, but the authors might want to mention again in discussion what they can expect for fate outcomes in a more realistic scenario, i.e. one where the cells are stimulated by a virus.

Line 62-65: I am not sure if the statement is correct for the Shaffer et al. – Is there evidence to show that the cells are epigenetically reprogrammed?

Line 104: how strong is the correlation between IRF3 and IRF7 translocation?

Lines 173-176: Hard to understand what the authors are trying to say here. Can the authors clarify what they mean?

Lines 190: Is 7h enough to get autocrine and paracrine signaling feedback loops?

I think the authors should state that their cell-cycle experiments are "inferred" experiments as they do not directly test this experimentally. If they wanted to make a more concrete statement, they may want to consider experiments using fucci reporters or using a CellTrace, although these experiments are beyond the scope of the paper, but the language used in the paper can be qualified.

Figure 3G the effect is barely there, so not sure what the authors mean by "significantly" in line 206. If they meant statistically significantly, then state that. As a side comment, I am not sure if this measurement is necessarily adding much to their inferred analysis on cell-cycle, and is rather distracting in the main text.

Comment: I liked the paragraphs and discussions on stochasticity and determinism in paragraph from line 214.

I think the way paragraphs of line 230 and line 243 are structured, they could go hand in hand instead of one after the other. Otherwise it gets a bit confusing as to what is analytical and what is experimentally observed.

Line 256: again, the authors say "heritable fates stated in literature" but largely only cite one paper throughout their manuscript.

I find the results from figure 4G confusing to understand and interpret in the greater context of the paper. I do appreciate the authors clearly stating themselves that the results are a bit puzzling but their plausible explanations are very speculative. Can the authors expand on this observation a bit more and leave open questions of interest to future studies?

Lines 304: I like that the authors clearly specify here that their way of measuring generations is only an indication.

Why did the authors start with 20 or 50 cells instead of one cell and stop at 6 generations? (an explanation is sufficient)

What does dark and light color distinctions in 6G mean? Do the cells change their form? Or is there a separate meaning to it?

The results from the density experiments are interesting and I like their attempt to use an ODE model to rationalize their results with a time/density dependent k_on_ parameter they define. However, for the authors to call it "quorum sensing" they should show a cell affecting its neighbors directly, rather than just from output measurements. Otherwise the authors may want to consider revising their language to call their results as simply "Density dependent".

For density experiments, have the authors considered an experimental design where instead of letting cells become dense (and have time to interact), just take sparsely plated cells in a larger well-plate and passage them into a small well plate, and add the polyIC right away? Will the authors see the density effect in this case?

Line 451: I do not understand what the authors are trying to convey here.

In general, I recommend that authors closely go over their figures and add details on colors, captions, and images in the revised manuscript.

*Reviewer #3 (Recommendations for the authors):*

1. In Figure 2I, it would be helpful to see quantification of the localization of the IFN response triggered by the early responders. From the one representative image, it is difficult to ascertain whether all of the majority of STAT1/STAT2 responding cells are in these clusters.

---

## [Author Response]

Essential revisions:Overall, the Reviewers agreed this is a well-designed, well-controlled, and timely study that presents important findings, which add to a growing body of evidence reporting heritable cell states in guiding cell fate choices. The experiments are well executed and clear, supporting the general claim that IFN-I early responder events are non-stochastic. However, the Reviewers were also in agreement about the limitations of the findings, specifically the generalizability of the finding and the link to epigenetic regulation, and shared concerns that some of the claims and terminology were not supported by the experiments. Finally, the Reviewers all raised important points about enhancing the clarity of the manuscript. The authors are expected to address the following points in their revision:1) Reviewers 1 and 3 both raised questions on the generalizability of the finding to other cell types, experimental systems, contexts. The authors are expected to address this comment either experimentally or, at the least by, by pointing to other relevant work.

Regarding your first point on the generalizability of the findings to other cell types, experimental systems, and contexts, we have revised the manuscript and added more examples from other studies (beyond IFN-Is) throughout. We agree it is important to report those, to provide more solid grounds on which our results are based. It is great to see that the reviewers themselves provided with very relevant literature to address the generalizability. Accordingly, we have added the following statements and literature:

On first responders:

“In short, IFN-I responses are elicited by fractions of so-called first responding cells, also referred to as ‘precocious cells’ or ‘early responding cells’, which start the initial IFN-I production upon viral detection, both validated in vitro, in vivo, and across cell types (Bauer et al., 2016; Hjorton et al., 2020; Patil et al., 2015; Shalek et al., 2014; Van Eyndhoven et al., 2021a; Wimmers et al., 2018).”

On transient heritability:

“Transient heritability refers to heritable epigenetic profiles [e.g., profiles encoding cellular fates for the production IFN-Is] that only transfer over a couple of generations, as observed across cell types and systems including cancer drug resistance (Shaffer et al., 2020), cancer fitness (Fennell et al., 2022; Oren et al., 2021), NK cell memory (Rückert et al., 2022), HIV reactivation in T cells (Lu et al., 2021), epithelial immunity (Clark et al., 2021), and trained immunity (Katzmarski et al., 2021).”

On quorum sensing:

“Besides a growing body of evidence on the role of transient heritable fates dictating cellular behaviors, the effects of population density, often referred to as quorum sensing, are getting more established for immune (signaling) systems (Antonioli et al., 2019; Polonsky et al., 2018; Van Eyndhoven and Tel, 2022). […] In essence, quorum sensing can be considered a phenomenon in which autocrine cells determine their population density based on cells engaging in neighbor communication, but without self-communication (Doğaner et al., 2016; Van Eyndhoven and Tel, 2022). Especially in the presence of other competitive decision makers [i.e., cytokine consumers and producers], it is critical for individual cells to assess cellular density, and act accordingly (Oyler-Yaniv et al., 2017).“

2) Reviewers 1 and 2 both raised important questions on the contribution of epigenetic regulation on first responder fate, specifically the lack of mechanistic links to DNA methylation. Suggestions for experiments are made below to further verify that epigenetic regulation modulates responder frequency.

Regarding the second point on the contribution of epigenetic regulation on first responder fates, we agree that the experiments performed for the draft manuscript were insufficient to provide a solid foundation to our statements. To improve, we have added additional experiments using a different epigenetic drug, as suggested by the reviewers, providing additional proof for the role of epigenetics in determining first responder fates. Besides, we added more nuance to our conclusions:

“Indeed, cells treated with DNMTis showed higher percentages of first responding cells, arguing that the cellular decision to become a responder is, at least partly, epigenetically regulated (Figure 4f). Also the treatment with varying dosages and durations of Trichostatin A, an histone deacetylase inhibitor (HDACi), increased the number of responding cells (Supplementary Figure 5).”

“In contrast, we currently hypothesize that first responders are predetermined, meaning that their epigenetic profile is driving their responder fate.”

Additionally, we went back to the literature to provide the reader with additional hypotheses and references addressing the proposed mechanisms underlying the phenomena observed. With that, we sincerely hope we have met the reviewer’s requirements, as our current, rather limited technological recourses and budget do not allow to further dive into this, unfortunately. Obviously, in future studies we want to further examine the exact (epigenetic) mechanisms underlying our findings, but as for now it is unfortunately beyond our reach. We hope you and the reviewers will understand.

The following hypotheses and literature were added to the revised manuscript to meet the requirement of strengthening the mechanistic foundation on the epigenetics driving our observed findings:

“The rather long timescales of switching from responders to non-responders, and the other way around, imply epigenetic mechanisms at play, and indeed, prior work has indicated an important role for epigenetics dictating IFN-I response dynamics (reviewed in (Barrat et al., 2019)).”

“Both methylation and histone acetylation have been suggested in dictating transient heritable cellular fates (Clark et al., 2021; Lu et al., 2021; Shaffer et al., 2020).”

3) Reviewers 1 and 2 raised relevant points on the limitations of the modeling and fitting which should discussed. Additionally, Reviewer 2 has suggested the authors temper their claims of "quorum sensing" and instead consider revising their language to call their result "density dependent".

Regarding the third point on the limitations of the modeling and fitting, we have added additional data on the young generation clones (generation 1, 2, 3, 4, 5). Moreover, we revised the model, which is now much more elegant and simple compared to the previous model and has been reported in literature before to explain similar phenomena on transient heritability. as the new model is not incorporating the density effect, we were able to thoroughly revise the use of the term “quorum sensing” throughout the manuscript.

4) Throughout their reviews, the Reviewers have made good suggestions for enhancing the clarity of the manuscript, specifically the need to provide clear definitions upfront, clarity in figures and legends, and elaboration on interpretations of their results (e.g. Figure 4G).

Regarding the fourth point on enhancing the clarity of the manuscript, we have incorporated all suggestions by the reviewers.

Reviewer #1 (Recommendations for the authors):Additional Suggested Experiments• Reproduce the generational variability and quorum sensing experiments in a distinct cell type or primary cells to confirm that transient heritability is a more general feature of IFN-I response and understand differences by cell lineage.• Further verify that epigenetic regulation (and not solely retrovirus activation) modulates responder frequency using either (1) an orthogonal epigenetic drug (e.g., histone deacetylase inhibitor) or (2) evaluating the addition of retroviruses into cells.• We propose the following extensions to help validate and improve the work, but they may be beyond the scope of available technologies that would be readily implemented:o Lineage tracing experiments using sequenceable barcodes to directly measure mother-daughter relationships and heritabilityo Measurement of DNA methylation, histone modification, or chromatin accessibility in first responders vs non-responders with and without DMNTi

We thank the reviewer for the useful and elegant additional suggested experiments. Regarding the reproducibility on generational variability and quorum sensing experiments in other cells (preferably primary cells), we are currently investigating that, proving the effects of quorum sensing driving responders in primary cells (unpublished data). The transiently heritability we are also currently exploring in primary cells. Regarding the verification on epigenetic regulation with an orthogonal epigenetic drug, we performed additional experiments using HDAC inhibitors.

Reviewer #2 (Recommendations for the authors):My comments are listed below, in no particular order:Could the authors more clearly state how the cutoff was defined for what IRF7 translocation ratio for responders versus non-responders? As it stands, it is unclear how the threshold was decided.

Regarding the IRF7 translocation cutoff, we clarified by adding the following additional information to the revised manuscript:

“Besides, the translocation ratio threshold for distinguishing responders from nonresponders was optimized per experiment, again based on visual validation.”

Figure 1E please label that X axis as IRF7 translocation ratio. Also, the current method of representation in 1E presents a high cognitive burden.The authors should consider using simpler visualization to make this point.Figure 2C legend can you define what the red dotted line and orange line signifies?Please add scale bars to all figures. Not a single image in the main text seemed to have scale bars.

All done.

The quantifications of mean intensities in Figures 3 and 4 are semi-quantitative at best. While I do not expect the authors to address this point experimentally (with a more quantitative method such as smFISH), they should highlight this as a limitation or add it as a discussion point somewhere.

Regarding the semi-quantitative methods we used, we added the following to the revised manuscript:

“Additionally, our utilized methodologies were only semi-quantitative. Perhaps, single molecule fluorescence in situ hybridization could provide an enhanced resolution to the cellular decision making observed in this study.”

The authors do not strictly identify "sister" or "twin" cells. Rather, they use neighboring cells as a proxy. Unless the authors actually trace sister cells in this study, this terminology should be avoided in the paper as it is misleading and incorrect.Along the same lines – how clear is it that cells do not move much for their neighboring cell analysis to be an accurate representation of progeny?

Regarding our word-choice on “sister” cells, we agree that our current methodology cannot fully prove that. However, we currently do not have the tools/methods available to implement those in the revised manuscript. Besides, we only reported our observation on responding “neighboring” cells as a proof of concept, leading to the hypothesis that responders are not purely stochastically regulated. We addressed the limitations accordingly:

“Also, after realizing that in the general experimental setup cells were seeded approximately 24 hours before imaging, allowing all cells to have divided once by the time of imaging, the appearance of responding neighboring cells could be further explained and quantified. Accordingly, we can assume that two responding neighboring cells are sister cells, although our current methodology lacks the ability to fully prove that.”

The authors have ignored several seminal studies when inspiring the idea of transient cell states guiding fate decisions. The authors are encouraged to read and mention some of these early studies (Spencer et al. 2009, Gupta et al. 2011, Sharma et al. 2010, and on LD: Fiddler and Kripke 1977) and some more recent ones on heritable fluctuations (Oren et al. 2021, Fennell et al. 2022, Quinn et al. 2022).

Thank you for providing such relevant references, to which we were unfamiliar. We have incorporated them, providing more nuance to our conclusions, for example:

“Perhaps, the transient heritability of epigenetic profiles is rather small, while stochastic protein states are also transmitted from mother to daughter cells, driving responder fates as well (Spencer et al., 2009).”

In the introduction, some of the terms are poorly defined or used without defining. One example is "multilayered stochasticity". Another example is contrasting stochasticity with determinism, which especially in the introduction is a bit misleading. The authors do clarify the nuance in the Results section briefly (lines 215-227) but the Introduction warrants a more nuanced discussion of these deep conceptual ideas. For example, even such heritable fluctuations, in principle, can arise from stochastic interactions (Schuh et al., 2020).

We agree that the introduction must provide all required (additional) explanation to the terms used throughout the manuscript. We have provided that accordingly:

On first responders:

“In short, IFN-I responses are elicited by fractions of so-called first responding cells, also referred to as ‘precocious cells’ or ‘early responding cells’, which start the initial IFN-I production upon viral detection, both validated in vitro, in vivo, and across cell types (Bauer et al., 2016; Hjorton et al., 2020; Patil et al., 2015; Shalek et al., 2014; Van Eyndhoven et al., 2021a; Wimmers et al., 2018).”

On transient heritability:

“Transient heritability refers to heritable epigenetic profiles [e.g., profiles encoding cellular fates for the production IFN-Is] that only transfer over a couple of generations, as observed across cell types and systems including cancer drug resistance (Shaffer et al., 2020), cancer fitness (Fennell et al., 2022; Oren et al., 2021), NK cell memory (Rückert et al., 2022), HIV reactivation in T cells (Lu et al., 2021), epithelial immunity (Clark et al., 2021), and trained immunity (Katzmarski et al., 2021).”

On quorum sensing:

“Besides a growing body of evidence on the role of transient heritable fates dictating cellular behaviors, the effects of population density, often referred to as quorum sensing, are getting more established for immune (signaling) systems (Antonioli et al., 2019; Polonsky et al., 2018; Van Eyndhoven and Tel, 2022). On top of the intrinsic features characterized by stochasticity and determinism, individual immune cells can communicate in various ways to elicit appropriate systemic immune responses. Typically, cytokine-mediated communication is categorized into two types: autocrine and paracrine signaling. Autocrine signaling is defined by cells secreting signaling molecules while simultaneously expressing the cognate receptor. Paracrine signaling is defined by cells either secreting signaling molecules without expressing the cognate receptor, or cells expressing the receptor without secreting the molecule. In essence, quorum sensing can be considered a phenomenon in which autocrine cells determine their population density based on cells engaging in neighbor communication, but without self-communication (Doğaner et al., 2016; Van Eyndhoven and Tel, 2022). Especially in the presence of other competitive decision makers [i.e., cytokine consumers and producers], it is critical for individual cells to assess cellular density, and act accordingly (Oyler-Yaniv et al., 2017).“

Besides, we have moved the detailed information on distinguishing stochasticity from determinism from the Results section to the introduction.

While the results with the DNMTi are quite interesting, a more detailed discussion and rationale of why demethylation was specifically chosen as a method to modulate the percentage of first responders would be helpful to put these results in context.

A more detailed discussion and rationale of why demethylation was specifically chosen as a method to modulate the percentage of first responders is provided in the revised manuscript:

“Continuing the hypotheses of transiently heritable cell fates stated in literature, we investigated the manipulation of cellular decision-making by altering the cells’ epigenetic profile, thereby altering any potential predispositioning towards becoming a first responder. Both methylation and histone acetylation have been suggested in dictating transient heritable cellular fates (Clark et al., 2021; Lu et al., 2021; Shaffer et al., 2020).”

“The rather long timescales of switching from responders to non-responders, and the other way around, imply epigenetic mechanisms at play, and indeed, prior work has indicated an important role for epigenetics dictating IFN-I response dynamics (reviewed in (Barrat et al., 2019)).”

In the experiments with DNMTi, the authors can clarify if the DNMTi was continuously provided or media was replaced before infection? In general, again, I do not expect the authors to do these experiments necessarily, but it would be good to comment on or hypothesize how their results may look like for pre-treatment, co-treatment (i.e. DNMTi added right when polyIC is added), or both for DNMTi.

Clarification on the methods for DNMTi incubation is provided in the revised manuscript:

“Therefore, cells were first incubated with DNA methyltransferase inhibitor (DNMTi) 5-Aza-2′-deoxycytidine (5-azadC) 24 hours prior transfection (Figure 4e). As indicated in the experimental schematic, the drugs were only administered pre-transfection.”

Have the authors considered an experiment where they sort the high responders and see if they lose this ability over time? Even if the authors do not perform the experiment, I am curious what the authors think of this design.

Regarding the point raised on whether first responders lose their ability to initiate IFN-I production upon a second stimulation (over time) was addressed in Van Eyndhoven et al., 2021 (PMID: 33995415).

It is outside the scope of this study experimentally, but the authors might want to mention again in discussion what they can expect for fate outcomes in a more realistic scenario, i.e. one where the cells are stimulated by a virus.

Excellent point! We are happy to share our thoughts on how we see these result translate to an in vivo situation:

“However, we do speculate that, in vivo, quorum sensing triggers collective and coordinated actions, as observed in dendritic cells activated in the draining lymph node, in vivo (Bardou et al., 2021). Intuitively, it makes sense that cell density plays a major role in cytokine-mediated responses, while the same antiviral effects are achieved by a high responding fraction of a small population of potential producers, as compared to a low fraction of a large population of potential producers (Van Eyndhoven and Tel, 2022).”

Line 62-65: I am not sure if the statement is correct for the Shaffer et al. – Is there evidence to show that the cells are epigenetically reprogrammed?

Verified.

Line 104: how strong is the correlation between IRF3 and IRF7 translocation?

Unfortunately, the quantification on the correlation between IRF3 and IRF7 has not been reported by Rand et al.

Lines 173-176: Hard to understand what the authors are trying to say here. Can the authors clarify what they mean?

Clarified.

Lines 190: Is 7h enough to get autocrine and paracrine signaling feedback loops?

According to the results reported by Rand et al., 7h post activation is sufficient to start autocrine and paracrine feedback loops, while the first IRF7 translocation starts around 5/6 hours post activation.

I think the authors should state that their cell-cycle experiments are "inferred" experiments as they do not directly test this experimentally. If they wanted to make a more concrete statement, they may want to consider experiments using fucci reporters or using a CellTrace, although these experiments are beyond the scope of the paper, but the language used in the paper can be qualified.

We agree that there are more elegant ways to validate cell-cycle arrest, however, using quantified cell numbers and literate to back our statements up, we still believe we can draw conclusions from our cell-cycle experiments. However, we agree we should be careful. Therefore, we added more nuance to the text:

“To explore the effect of cell cycle state on first responding cells, we aimed to synchronize the cells using serum starvation for 24 hours (Figure 3d). This approach induces a cell cycle arrest, halting cells in the G0/G1 phase, thereby synchronizing the whole population (Chen et al., 2012).”

Figure 3G the effect is barely there, so not sure what the authors mean by "significantly" in line 206. If they meant statistically significantly, then state that. As a side comment, I am not sure if this measurement is necessarily adding much to their inferred analysis on cell-cycle, and is rather distracting in the main text.

We indeed meant statistically significantly, which we clarified in the text of the revised manuscript. We also agree that the effect in Figure 3G seems only minorly, but we believe it provides an additional piece of evidence that our cells were properly synchronized by halting their cell cycle.

Comment: I liked the paragraphs and discussions on stochasticity and determinism in paragraph from line 214.

Good! Though, we moved parts to the introduction upon the request of other reviewers.

I think the way paragraphs of line 230 and line 243 are structured, they could go hand in hand instead of one after the other. Otherwise it gets a bit confusing as to what is analytical and what is experimentally observed.Line 256: again, the authors say "heritable fates stated in literature" but largely only cite one paper throughout their manuscript.

We added more literature to the revised manuscript to provide the readers with more references on (transiently) heritable fates.

I find the results from figure 4G confusing to understand and interpret in the greater context of the paper. I do appreciate the authors clearly stating themselves that the results are a bit puzzling but their plausible explanations are very speculative. Can the authors expand on this observation a bit more and leave open questions of interest to future studies?

We agree our explanation on the results from Figure 4G are currently rather speculative, and our current methodology cannot provide additional answers/explanation. We removed the open questions to avoid unnececary confusion.

Lines 304: I like that the authors clearly specify here that their way of measuring generations is only an indication.

Good.

Why did the authors start with 20 or 50 cells instead of one cell and stop at 6 generations? (an explanation is sufficient)

We provided additional explanation in the revised manuscript:

“Of note, in this experimental setting, the generation number is only an indication of the amount of cellular divisions that the clone has undertaken, rather than a determinantal factor, as cells do not remain synchronized over multiple generations. Performing a limited dilution for all clones would have been more elegant, thought, it was practically too challenging to seed single cells on coverslips to be imaged from the day afterwards. Instead, upon low cell seeding, there is still enough empty space surrounding the clusters of cells to determine which cells originated from a single cell (Supplementary Figure 7A-D, and Supplementary Figure 8A-D).”

What does dark and light color distinctions in 6G mean? Do the cells change their form? Or is there a separate meaning to it?

We used different colors to distinguish the unstimulated state from the stimulated state of a cell, driving responsiveness. However, we removed Figure 6G as our model has changed drastically.

The results from the density experiments are interesting and I like their attempt to use an ODE model to rationalize their results with a time/density dependent k_on_ parameter they define. However, for the authors to call it "quorum sensing" they should show a cell affecting its neighbors directly, rather than just from output measurements. Otherwise the authors may want to consider revising their language to call their results as simply "Density dependent".

After we obtained additional data point for the early generation clones, we did no longer incorporate a density-dependent k_on_, while the data could be very elegantly fitted with a model that only considers transient heritability. Additionally, we revised our choice of words regarding quorum sensing effects versus density-dependent effects throughout the manuscript, as this was also brought up by other reviewers.

For density experiments, have the authors considered an experimental design where instead of letting cells become dense (and have time to interact), just take sparsely plated cells in a larger well-plate and passage them into a small well plate, and add the polyIC right away? Will the authors see the density effect in this case?

This is a very good suggestion. We have not tested that yet, but we are planning to perform similar experiments in future studies.

Line 451: I do not understand what the authors are trying to convey here.In general, I recommend that authors closely go over their figures and add details on colors, captions, and images in the revised manuscript.

We agree that this was a confusion sentence, which we removed in the revised manuscript. Finally, also upon the request by the other reviewers, we thoroughly revised the figures and captions for an enhance clarity of the manuscript.

Reviewer #3 (Recommendations for the authors):1. In Figure 2I, it would be helpful to see quantification of the localization of the IFN response triggered by the early responders. From the one representative image, it is difficult to ascertain whether all of the majority of STAT1/STAT2 responding cells are in these clusters.

Regarding the statistics for STAT_1/2_ translocation data, we only showed this data as a proof of concept, indicating that cells produce IFN-I upon activation. In a different manuscript that is currently in preparation, we solely focus on the characterization of STAT_1/2_ translocation dynamics. Therefore, we consider the proposed quantification beyond the scope of this manuscript, while we aimed to focus on the early IFN-I response dynamics only.